# Aero-elastic analysis of wind turbines under turbulent inflow conditions

Giorgia Guma[1], Galih Bangga[1], Thorsten Lutz[1], and Edwald Krämer[1]

[1]Institute of Aerodynamics and Gas Dynamics, University of Stuttgart, Pfaffenwaldring 21, 70569 Stuttgart, Germany

**Correspondence:** Giorgia Guma (guma@iag.uni-stuttgart.de)

**Abstract.** The aero-elastic response of a 2 MW NM80 turbine with a rotor diameter of $80m$ and the interaction phenomena are investigated by the use of a high-fidelity model. A time-accurate unsteady fluid-structure interaction (FSI) coupling is used between a computational fluid dynamics (CFD) code for the aerodynamic response and a multi-body simulation (MBS) code for the structural response. Different CFD models of the same turbine with increasing complexity and technical details are coupled to the same MBS model in order to identify the impact of the different modeling approaches. The influence of the blade and tower flexibility and of the inflow turbulence is analyzed starting from a specific case of the DANAERO experiment, where a comparison with experimental data is given. A wider range of uniform inflow velocities are investigated by the use of BEM as aerodynamic model. Lastly a fatigue analysis is performed from load signals in order to identify the most damaging load cycles and the fatigue ratio between the different models, showing that a highly turbulent inflow has a larger impact than flexibility, when low inflow velocities are considered. The results without the injection of turbulence are also compared and discussed to the one provided by the BEM code AeroDyn.

## 1 Introduction

The current design trend of wind turbines is leading to rotor diameters getting larger and larger, but they have to be light in order to decrease the cost of wind power generations in terms of leveling energy costs ($\$/kWh$) and make it a competitive resource in comparison to other electric generation systems. A lot of research is done to investigate materials and construction techniques in order to allow lighter designs with the consequence that the rotor blades are becoming more and more flexible, which leads to large deformations with associated non-stationary loads and oscillations, resulting in unexpected changes in performances or even flutter, if the damping is negative. Additionally, large rotor wind turbines are in reality subjected to diverse inflow conditions, such as shear, turbulence and complex terrain, leading to higher load fluctuations. Moreover, the aerolastic instabilities strongly affect the operational life of wind turbines (Hansen et al. (2006)). Most of the available simulation tools for wind turbines aeroelasticity are based on engineering models like BEM for the aerodynamics and 1D MBS for the structural response, like for example in Riziotis et al. (2008) and Jeong et al. (2011). These models are cheap but rely on different correction models to take unsteadiness and 3D effects into account (Madsen et al. (2012)). In recent years, high-fidelity FSI has been frequently used for wind turbine applications. Sayed et al. (2016) implemented a coupling of the CFD solver FLOWer to the CSD (Computational Structure Dynamics) solver Carat++, where only the blades have been coupled either to a 1D

beam or a 2D shell structural model. Yu et al. (2014) used a loose CFD-CSD coupling with an incompressible CFD solver and non-linear Euler-Bernoulli beam elements for the structure in order to investigate the aeroelastic response of the generic NREL 5 MW rotor. The communication in this case was only once per revolution. The same turbine was also used by Bazilevs et al. (2011b) and Hsu et al. (2012) by means of FSI between a low-order Arbitrary Lagrangian-Eulerian Variational Multi Scale (ALE-VMS) flow solver and a Non-Uniform Rational Basis Spline (NURBS) based structural solver. For the same turbine, Heinz et al. (2016) compared the coupling of the flow solver Ellypsys3D with the aeroelastic solver HAWC2 to the BEM results of HAWC2 alone. While he considered uniform inflow, Li et al. (2017) additionally considered a turbulent inflow synthetically generated by the use of a Mann box (Mann (1994)). Dose et al. (2018) presented a method to couple the flow solver OpenFOAM to the FEM-based beam solver BeamFOAM. A CFD-MBS coupling between the URANS solver TURNS and the MBS solver MBDyn was used by Masarati et al. (2011) to investigate the NREL Phase VI rotor.

Wind turbines are especially susceptible to fatigue damage, due to the oscillating characteristic of the affecting loads. Fatigue analysis are normally performed by manufacturers for certification purposes, and therefore they are mostly BEM-based. In the EU-project AVATAR (Schepers (2016)) it was shown that BEM-based calculations against high fidelity calculations led to a 15% error in the computation of fatigue. This error motivated the TKI WoZ VortexLoads project (Boorsma et al. (2019)), where starting from turbulent inflow conditions BEM based and CFD based calculation have been compared with each other and to experimental results.

Within the scope of the present study, a highly accurate CFD-based aeroelastic model of a 2MW wind turbine was created and applied to study the unsteady load characteristics. The objective was to identify the impact of the modelling of the individual turbine components and the occurring interactions on the transient loads. To achieve this goal numerical models of successively increasing complexity are introduced. Starting from a one-third model of the blade in uniform inflow, over a complete rotor up to a complete flexible turbine in turbulent inflow, the transient loads were analyzed and compared. The aim was to analyze the main drivers for the load fluctuations and the Damage Equivalent Loading (DEL) using highly accurate models. R1:G1b The different CFD configurations have been analyzed in detail because their computational costs vary enormously. It is therefore of interest, especially for the industry to know limitations and differences within the high-fidelity approaches. For the uniform inflow case, a comparison with BEM based calculations is given and two additional inflow conditions are computed, because of its cheapness, in order to determinate the generalization level of the results. The ability of BEM of predicting reliable fatigue values changing the computational settings is discussed. In section 2 of this paper, the high-fidelity framework (as presented in Klein et al. (2018)) is described for fluid-structure interaction coupled simulations on the NM80 2MW wind turbine rotor, also known as DANAERO rotor, (DANAERO). The inflow conditions and setup for the different cases are described. Furthermore, the BEM model of the turbine is described with its validation, basing on the usage of 3D CFD polars in order to ensure consistency with the high-fidelity model. In section 3, the aeroelastic response of the reference turbine is shown and the difference between the modelling approaches is exposed. Lastly, DEL calculation is performed in post processing of the different simulations, using two different time varying input variables.

## 2 Methodology

### 2.1 DANAERO wind turbine

The DANAERO wind turbine rotor is used for this paper. This is the reference wind turbine in the IEA Task 29 IV, also known as MEXNEXT IV, (IEA Task 29). In this project different, institutions and universities around the world compare their own codes and approaches, using them for the calculations planned into different subtasks of the same project. The results are not only compared to each other, but also to experimental results provided by the DANAERO experiment (Madsen et al. (2010)). The experiment were conducted between 2007-2010 in cooperation between the Technical University of Denmark and the industrial partners Vestas, Siemens LM and DONG Energy, and then post processed and calibrated in the follow up project DANAEROII, (Troldborg et al. (2013)). In this way it is possible not only to understand limitations and problematics of the different approaches, but also to improve them. The turbine has a rotor diameter of around 80 m, a tilt angle of 5 degrees and around 1.4 m prebend. Hub, nacelle and tower have been modelled within the present study as cylinders, based on the available diameter distribution provided in the structural model.

### 2.2 CFD model and inflow conditions

The simulations are performed with the CFD code FLOWer (Raddatz (2009)). Firstly developed at the German Aerospace Center (DLR), FLOWer is now since many years expanded at the Institute of Aerodynamic and Gas Dynamic (IAG) for helicopter and wind turbine applications. It is a URANS and DES finite volume solver for structured meshes. The present simulations are run using the Shear-Stress-Transport (SST) k-omega model according to Menter (Menter (1994)), using a fully turbulent boundary layer. Two different spatial discretization schemes are available, a second order central cell-centered Jameson-Schmidt-Turkel (JST) ( (Jameson et al.(1981)) and a fifth order weighted essentially non-oscillatory (WENO) (Kowarsch et al. (2013)) scheme. The second one is applied in the present study on the background mesh in order to reduce the dissipation of the vortices. The time-stepping scheme is an artificial 5-stage Runge-Kutta scheme and multi-grid level 3 is applied to accelerate the convergence of the solution. The time integration scheme is an implicit procedure called dual-time stepping where at the beginning of each timestep $t$ an estimation of the solution is guessed. The closer this is to the final value, the smaller the necessary number of inner iterations to reach convergence. Independent grids need to be created for each single component, combined and overlapped by the use of the Chimera technique.

The CFD model of the blade is created from the provided CAD file, where a "water tight" outer surface is extracted. For hub, nacelle and tower, surface databases are recreated (cylinder-based) from provided geometrical properties. Meshes are generated by the use of the commercial software Pointwise in combination with in-house scripts. All components have been meshed ensuring $y^+ \leq 1$ in the boundary layer region. The blades are meshed in an O-mesh topology with 257 points over the the profile and 201 points in radial direction, for a total of around 9 Mio cells for each blade. The background mesh consists of hanging grid nodes in which the component meshes are embedded with the Chimera technique. Three different CFD models have been created for the turbine, with increasing fidelity:

1. One-third model (BMU) of the rotor (only one blade) suited for uniform inflow conditions;

2. Full model of the turbine (FMU) including nacelle and tower suited for uniform inflow conditions;

3. Full model of the turbine (FMT) including nacelle and tower suited for turbulent inflow conditions;

The differences between the three models consist in the background that were used. Model 1 has no ground, because it is just a 120° model of the turbine. Model 2 has no friction on the ground in order to avoid the generation of a wind profile. Finally, model 3 has friction on the ground in order to consequently propagate the sheared turbulent inflow and is much more expensive in comparison to case 2 (87 Mio cells against 58 Mio), because an additional refinement is added upwind where the turbulence is injected, and different boundary conditions need to be applied in order to ensure a correct propagation of the turbulence. The 120° model is much cheaper than the other two, because it uses the periodic characteristic of a 3-bladed wind turbine, but of course it considers neither tilt angle nor tower influence. The different boundary conditions and CFD models are depicted in fig. 1. In the following the meaning of the different boundary conditions is clarified:

– NAVIER-STOKES and EULER wall represent the ground with and without friction, respectively;

– FARFIELD represents the uniform inflow boundary condition;

– PERIODIC/PERIODIC ROT represent the symmetrical boundary condition for the full and 120° model, respectively;

– GUST is the Dirichlet boundary condition, by which arbitrary unsteady inflow can be applied;

– PRESSURE OUTLET defines the outflow based on pressure;

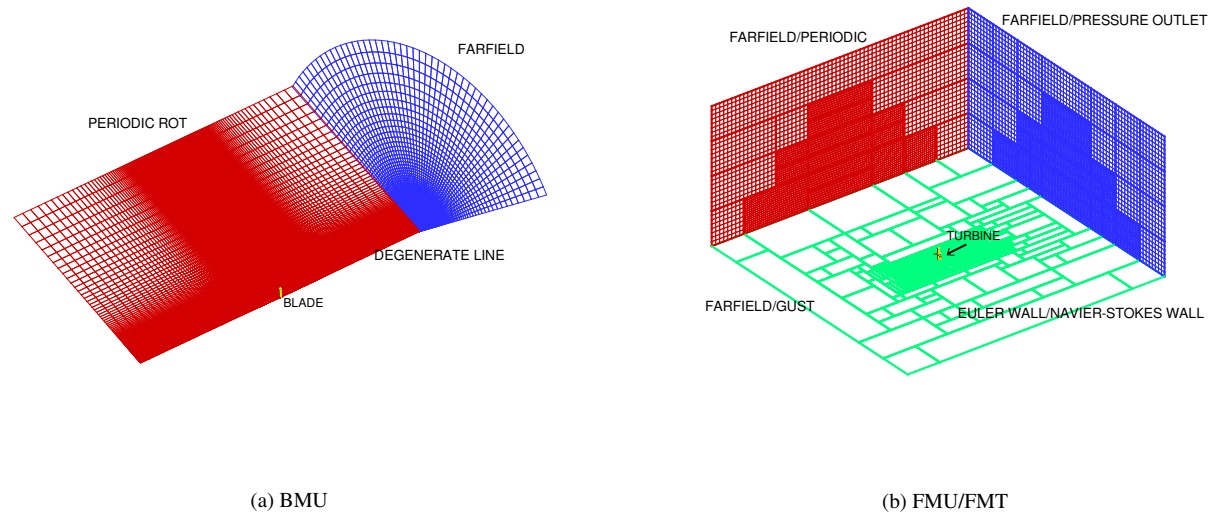

(a) BMU

(b) FMU/FMT

**Figure 1.** Details of the meshes and boundary conditions for BMU and FMU/FMT

All simulations are run based on the conditions defined in the subtask 3.1 of the IEA task 29, see (IEA Task 29). Those require a rated inflow velocity of 6.1 $m/s$ in the uniform case. For FMT, synthetic turbulence is generated by the use of a Mann Box (Mann (1994)) and injected in the flowfield at a plane 4 diameters (4D) upstream from the tower bottom. This is added using a momentum source term as prescribed in Troldborg et al. (2014) and superimposed to the steady uniform inflow. The turbulence on this plane is updated every time step using Taylor's frozen turbulence hypothesis (Troldborg et al. (2014)). A Turbulence Intensity (TI) of 20%, a length scale of 0.59*hub height (according to the IEC standard normative 61400) and a stretching factor $\Gamma = 3.9$ to approximate the Kaimal spectral model (as prescribed in Kim et al. (2018)) are preset. A mesh refinement of the background is applied from the inflow plane in order to allow a better propagation of the turbulence. The effective TI at the rotor is usually lower than the one prescribed in the Mann box, because it decays for both physical and numerical reasons. From an empty box calculation with a TI of 6.8% a turbulence decay of around 14% was calculated, and therefore it is assumed for this case that the effective TI amounts to 17.2%. Sheared inflow is superimposed by the use of a power law with $\alpha =0.025$. Due to the low reference velocity considered during the DANAERO experiment, a really high TI was chosen in order to be able to identify distinctively the effects of a turbulent atmospheric boundary layer. DDES is used instead of URANS for the CFD solution, changing the boundary conditions accordingly.

## 2.3 MBS solver

### 2.3.1 Structural model

The multi-body dynamics (MBD) simulation code SIMPACK is used to simulate the structural dynamics of the turbine (as in Jassmann et al. (2014) and Luhmann et al. (2017)). The structural properties of the entire turbine have been modeled starting from the provided HAWC2 aeroelastic data. A multi-body system consists of rigid or flexible bodies interconnected by force and joint elements that impose the kinematic and dynamic constraints. Each body, represented by one or more markers, may then have three translational and rotational displacements as result of deformations and motion. The body motion is described by a set of Differential-Algebraic Equations (DAEs), a combination of differential motion equations and algebraic constraints. The blades are modeled as nonlinear SIMBEAM body types (three dimensional beam structures in SIMPACK, described by a node-based nonlinear finite differences approach). These have been discretized into 22 Timoschenko elements in radial direction, taking into consideration also gravitational and centrifugal forces. Structural damping is applied using the Rayleigh damping model with $\alpha = 0.025$ and $\beta = 0.014$. Due to its small expected deflections, the tower has been modeled as a linear SIMBEAM discretized into 25 Euler-Bernoulli elements, the hub has been modeled with 2 linear Euler-Bernoulli elements and the nacelle is modeled with one only rigid node, i.e. it can move but not deform, see fig. 2. Loads provided from the CFD are damped for the first 200 timesteps (equivalent to 200 Azimuth degrees) in order to avoid strong and fast deformations that can lead to numerical instabilities in the calculation. In order to validate the structural model, the natural frequencies of the singles blade and turbine are compared to the measured ones from Hansen et al. (2006) in table 1.

| Full Turbine Measured | Full Turbine Computed |
|---|---|
| 0.437 | 0.4812 |
| 0.444 | 0.4862 |
| 0.839 | 0.869 |
| 0.895 | 0.9201 |
| 0.955 | 0.9626 |
| 1.838 | 1.8758 |
| 1.853 | 1.912 |
| 2.135 | 2.5477 |
| 2.401 | 2.7265 |

| Single Blade Measured | Single Blade Computed |
|---|---|
| 1.01 | 0.938 |
| 1.91 | 1.884 |
| 2.96 | 2.687 |

**Table 1.** Comparison natural frequencies between the measured ones and the computed by SIMPACK: single blade on the left and full turbine on the right.

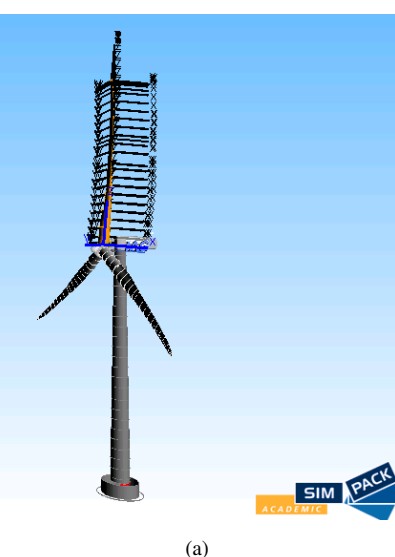

(a)

**Figure 2.** Visualization of the structural MBD model

## 2.4 BEM model

A simplified aerodynamic model based on Blade Element Momenutm (BEM) theory has been generated with the NREL code AeroDyn (Aerodyn (2005)). This has the advantage of beeing already incorporated in SIMPACK as additional module, and it can be therefore easily coupled to the structural model. In this case, the blade needs to be modeled aerodynamically with

145 as many nodes as structurally, i.e. 21 for each blade. Polars have been extracted from 3D CFD calculations in order to avoid the use of any tip or hub correction model and ensure as much consinstency as possible to the CFD calcualtions, as it was already shown in Guma et al. (2018). The 3D polars have been provided in a range of AOA between around $-30°$ to $+30°$ and have been extracted from the CFD solution using the RAV method (Rahimi et al. (2018)) and then extrapolated up to $-180°$ to $+180°$ using the Viterna method. Axial and tangential induction corrections have been taken into account. Tower

shadow effect has been taken into account depending on the computed case (single blade or full turbine). The comparison of the sectional loads per unit length in normal ($F_N$) and tangential ($F_T$) direction between BEM and CFD is depicted in fig. 3. In this case only one blade, with no tower shadow and rigid conditions has been taken into consideration, averaging the results of the three last revolutions. The curves show a good agreement, and therefore the BEM model of the turbine is validated. **R1:G3**
The discussion of limitations and capabilities of BEM under turbulent inflow conditions is out of the scope of this paper. This

aspect has been already addressed by Madsen et al. (2018), who compared BEM based simulations using the aerelastic tool HAWC2 to the high-fidelity code EllipSys3D and to experiments. A good agreement was found between the three, although CFD predicted an unforeseen stall in the inboard regions. In the present work only uniform inflow cases have been calculated using BEM as aerodynamic model of the turbine. The chosen setups are shown in table 2.

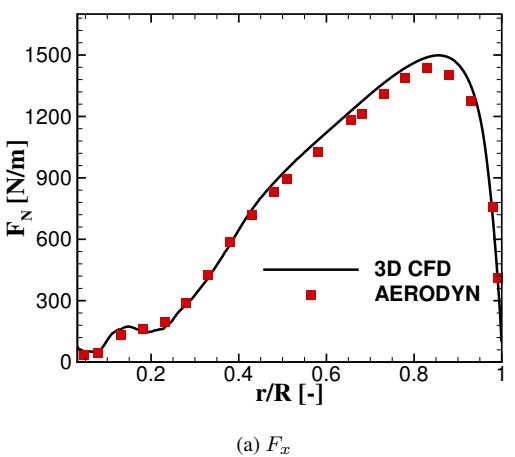
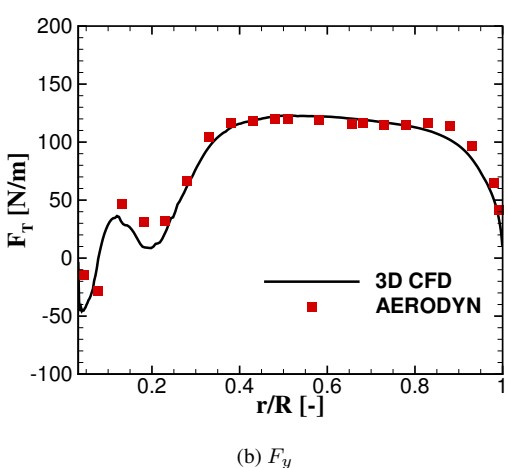

(a) $F_x$                                                 (b) $F_y$

**Figure 3.** Normal (on the left) and tangential (on the right) sectional load in comparison for a single rigid blade 3D CFD vs AeroDyn

| Inflow Velocity ($m/s$) | RPM | Pitch Angle (°) |
|---|---|---|
| 6.1 | 12.3 | 0.15 |
| 9.0 | 17.83 | 1.20 |
| 13.0 | 19.08 | 3.49 |

**Table 2.** Computed cases with uniform inflow in BEM. The first case is the one computed also with CFD.

## 2.5 FSI setup and computed cases

In order to allow the communication between FLOWer and SIMPACK, moving, undeformed and reference system markers need to be defined as prescribed in (Klein et al. (2018)). In the present study no controller is taken into account, that is why each simulation is conducted with a fixed rotational speed and pitch. These have been set according to the inflow velocity of 6.1 $m/s$, that is at the same time the chosen uniform inflow velocity and the average velocity at which the Mann box is generated. Even if a high TI is set, the resulting velocity is always far away from cut-off. Therefore, the controller would mainly change the RPM and not the pitch angle. The change in RPM has an influence on the full system natural frequencies (that is expected to be small), on the blade-tower passage frequency, and on the thrust. This would increase with the RPM and therefore the flapwise tip deformations. The used coupling algorithm is explicit, i.e. deformations and loads are exchanged only once per physical timestep. In particular, the loads at the end of the flow calculation timestep are used to calculate deformations that are applied to the subsequent step, see fig. 4. The chosen timestep in this case corresponds to 1 azimuthal degree. An already converged rigid simulation of the turbine that ran already for at least 10 revolutions is used as restart for the coupled simulation in order to speed up the calculation and save computational time. The DANAERO rotor has a high induction, therefore it takes many revolutions for the wake to fully develop and for the loads to stabilize. In order to save computational time, turbulence is injected and flexibility is activated, only after a cheaper simulation (FMU) reached a low residuum, a difference lower than 1% in the averaged loads and deformations between two revolutions, and a wake development long enough to avoid effects on the loads too.

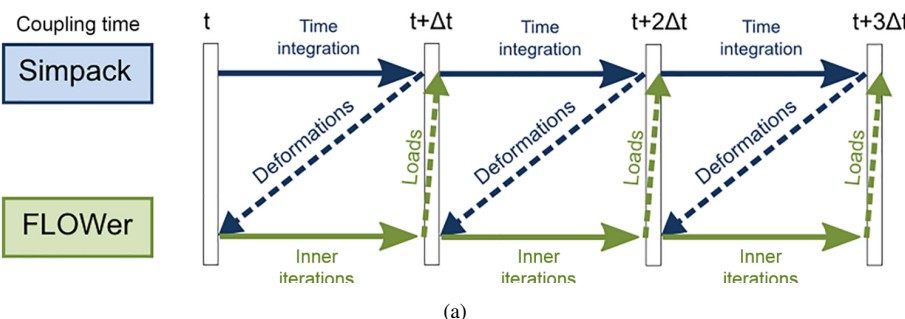

(a)

**Figure 4.** Explicit coupling strategy

For the BMU case it was sufficient to run the coupled simulation for only 6 further revolutions to achieve convergence and periodicity of the results. For the FMU, RMU and FMT at least 10 revolutions have been run, although periodicity cannot be reached in the FMT case, because the simulation time is much shorter than the length of the used Mann box. The elapsed time for the coupled simulations (starting from a rigid converged solution) varies from a minimum of 15 hours with 1632 processors for the BMU to a maximum of 48 hours with 4320 cores for the FMT case. All simulations are run on the SuperMUC-NG supercomputer at the Leibniz-Rechenzentrum in Munich.

All the CFD-MBD computed cases and differences can be seen in table 3. For each mentioned case a rigid and a coupled version is available, although RMU R (rigid) and FMU R (rigid) represent the same case.

| Case Name | Inflow Conditions | CFD Structures | Flexible Structures |
|---|---|---|---|
| BMU | uniform | one blade and 1/3 hub | blade |
| RMU | uniform | rotor, nacelle, tower | rotor |
| FMU | uniform | rotor, nacelle, tower | rotor, nacelle, tower |
| FMT | sheared turbulent inflow | rotor, nacelle, tower | rotor, nacelle, tower |

**Table 3.** Computed cases with inflow condition, CFD modeled structures and flexibility

## 2.6    Damage Equivalent Loading (DEL)

The DEL is a constant load that leads, when applied for a prescribed number of cycles, to the same damage as that caused by a time varying load over the same period. With this method, two or more signals can be compared in order to get insight into the fatigue loadings that blades are facing during normal operation. The approach is based on the S-N curves (stress vs number of cycles) of the material on a log-log scale, so that the material behavior is defined by the slope of a line. Additionally, a rainflow algorithm is applied to recognize the relative fatigue cycles in a load signal by filtering peeks and valleys. This algorithm allows

to estimate the amount of loads change depending on the amplitude of the cycle. In this way closed stress hysteresis cycles can be identified defining not only their amplitude, but also how often they appear. The consequent damage is, in fact, dependent on the combination of the last two factors. The used formulation in this paper is the one from Hendriks et al. (Hendrinks et al. (1995)) in which the different load signals are compared on a quantitative basis and using not only the range but also the mean of the load cycles. According to this method, the final expression of the DEL resulting from a prescribed signal is:

$$DEL = S_{r,eq} = \left( \sum_{i=1}^{n} \frac{\left( S_{r,i} * \frac{S_u - S_{m,eq}}{S_u - S_{m,i}} \right)^m}{Neq} \right)^{\frac{1}{m}} , \tag{1}$$

    where $n$ is the total number of cycles detected by the rainflow counting, $S_{r,i}$ is the amplitude of the $i-th$ cycle, $S_u$ is the ultimate load, $S_{m,i}$ is the mean value of the $ith$ cycle, $Neq$ is the number of cycles corresponding to DEL, $S_{m,eq}$ is the equivalent mean value of the cycle with amplitude DEL and, finally, $m$ is the slope of the S-N curve, considering a symmetric Goodman diagram with straight life lines.

$S_{r,i}$ and $S_{m,i}$ are direct output of the rainflow counting, meaning that they are an individual and inevitable characteristic of the spectrum itself. Differently, $Neq$, $S_u$, $S_{m,eq}$ and $m$ need to be chosen in advance. $S_u$ and $m$ are material dependent, where a log-log S-N curve is considered in order to have a straight line, respectively a constant $m$, while $S_u$ can be calculated in first

approximation as 5 times the maximum load in the provided spectrum. $Neq$ and $S_{m,eq}$ are user dependent. It is then clear that the absolute value computed by the DEL strongly depends on the choice of the constants, but as long as the same constants are considered, the DEL values are consistent within each other and, therefore, comparable.

## 3 Results

### 3.1 Aeroelastic effects

In this first section, the effects of aeroelasticity on the reference wind turbine are analyzed. The considered DANAERO experiment was performed at a low inflow velocity (6.1 $m/s$), that is why it is expected to have small deformations, and therefore especially a low tower effect. The used structural model is always the same, imposing opportunely the flexibility of the components as prescribed in table 3. This means that the calculation of gravitational and centrifugal forces, that is made directly in SIMPACK, is always taking the tilt angle into account, even in the BMU case.

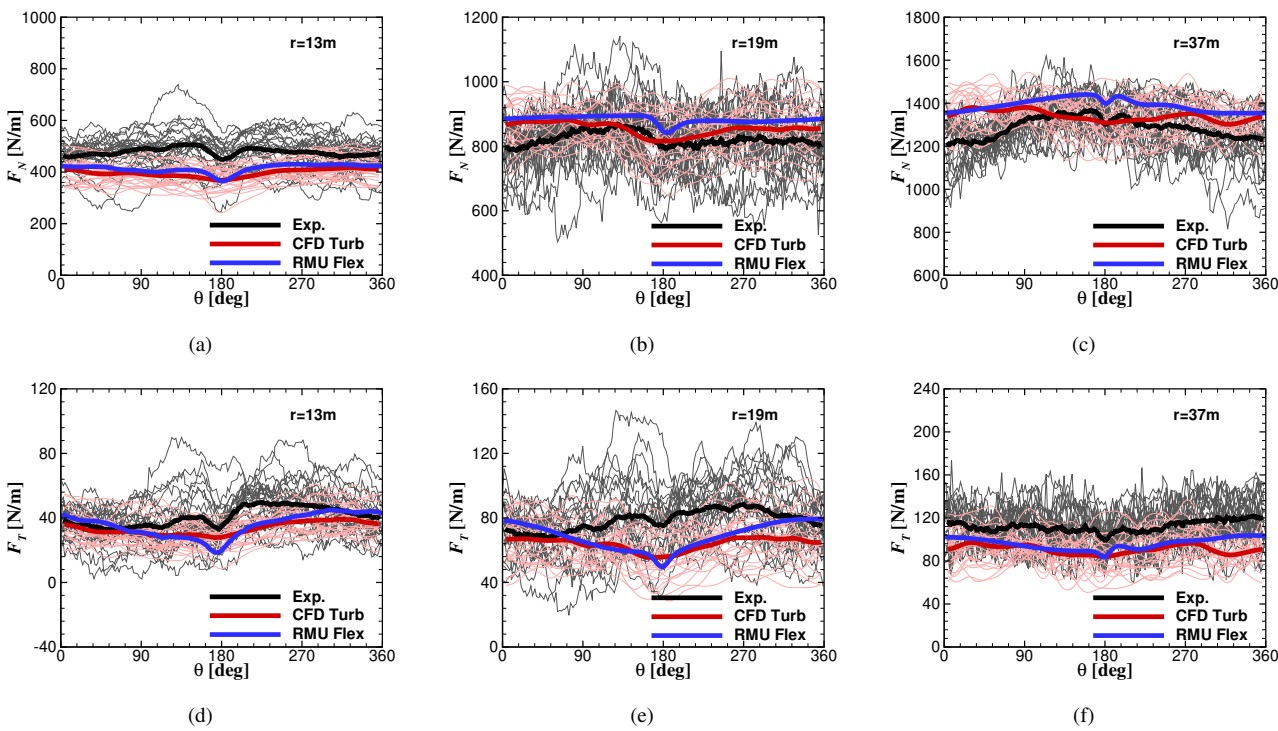

**Figure 5.** Comparison of experimental normal loads ($F_N$) in (a), (b), (c) and tangential loads($F_T$) in (d), (e), (f) for three different radial sections ($r = 13m$, $r = 19m$, $r = 37m$) over the blade. The blue line represents the full turbine with flexible blades. The red line represents a rigid rotor without tower but a turbulent inflow with the same TI as in the experiments. Grey and pink thin lines represent the data per revolution for the experiments and "CFD Turb", respectively.

As validation of the results, the sectional normal ($F_N$) and tangential ($F_T$) loads according to the chord length for 3 different radial positions in comparison to experiments are shown in fig. 5.

Results of different field tests have been considered and averaged (black line). As described in section 2.2, turbulence has been generated in a stochastic way, and therefore the experimental and simulation time series of each revolution are not directly comparable, but need to be averaged. For the validation, two different test cases have been compared: an entire CFD model of the turbine with flexible blades with uniform inflow conditions (RMU C, blue line) and an only rotor CFD model completely rigid but with an inflow turbulence comparable to the experiments (CFD Turb, red line). It can be seen that in the outside region, although a correct modeling of the inflow provides results closer to the experiment, the shape of the experimental curve is mostly good matched by the RMU C curve. In the hub region, the two modeling approaches do not show much difference from each other, although the flexible case gives slightly better results.

### 3.1.1 BMU vs RMU

The first considerations are made comparing BMU and RMU; the two differ from each other by the presence of a rigid tower and a tilt angle in the CFD model. Deformations in flap-wise, edge-wise and torsion direction of the tip of the blade can be seen in fig. 6. It can be noticed that, due to the inertia of the blade, the tip deformation starts its downturn by $180°$ but shows this local minimum with a delay of around $20°$ by $2.35\%$ of the rotor radius.

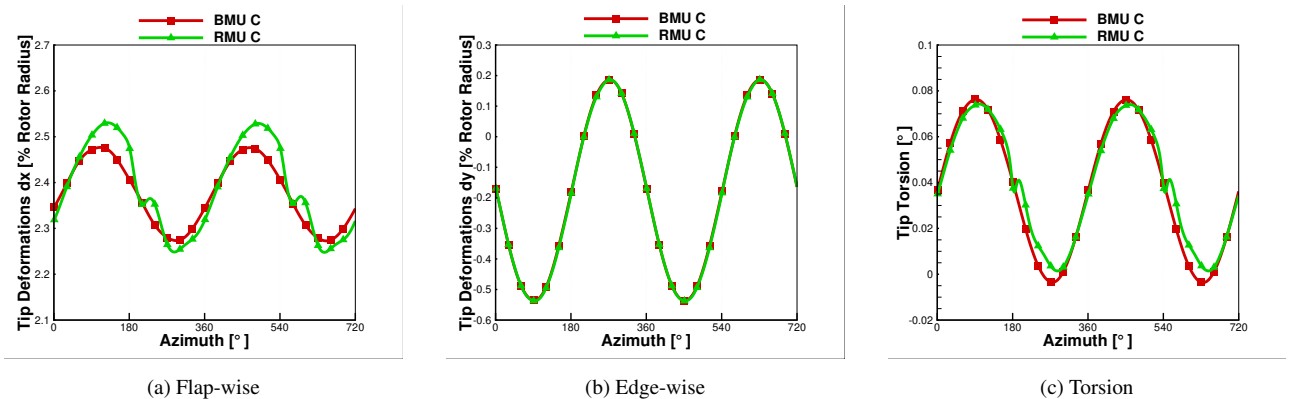

|        |        |        |
| ------ | ------ | ------ |
| (a) Flap-wise | (b) Edge-wise | (c) Torsion |

**Figure 6.** Tip deformations calculated with CFD at 6.1 $m/s$ in comparison: BMU coupled (C) vs RMU coupled (C). Out-of-plane deformation in (a), in-plane deformation in (b) and torsion in (c).

A clear sinusoidal trend can be seen in both cases, that leads to an oscillation of the tip deflection from around $2.3\%$ to $2.5\%$ of the blade radius for the BMU case, and from around $2.2\%$ to $2.5\%$ for the RMU case. The reason for this is the presence of the tilt angle ($5°$) that leads the gravitational and centrifugal forces to produce an oscillating deformation component in flap-wise direction. On the contrary, the aerodynamic contribution remains almost constant in time, with an oscillation smaller than $1\%$. As previously mentioned, the CFD model in BMU has no tilt, but the structural model does, that is why the resulting centrifugal and gravitational forces are accordingly affected. This leads to the oscillation in the response of BMU. This oscillation turns

out to be stronger than the blade-tower passage for RMU, therefore after the minimum due to the blade-tower interaction, there
is a recovery that immediately collapses in order to follow the sinusoidal trend. The difference in the maximum deflection
between BMU and RMU is 2.4% and is due to a higher oscillation of the affecting loads in the rigid version of RMU, as can
be seen in fig. 7, where the global thrust ($F_x$) and torque ($M_x$) in the rigid and coupled case on the blade are plotted.

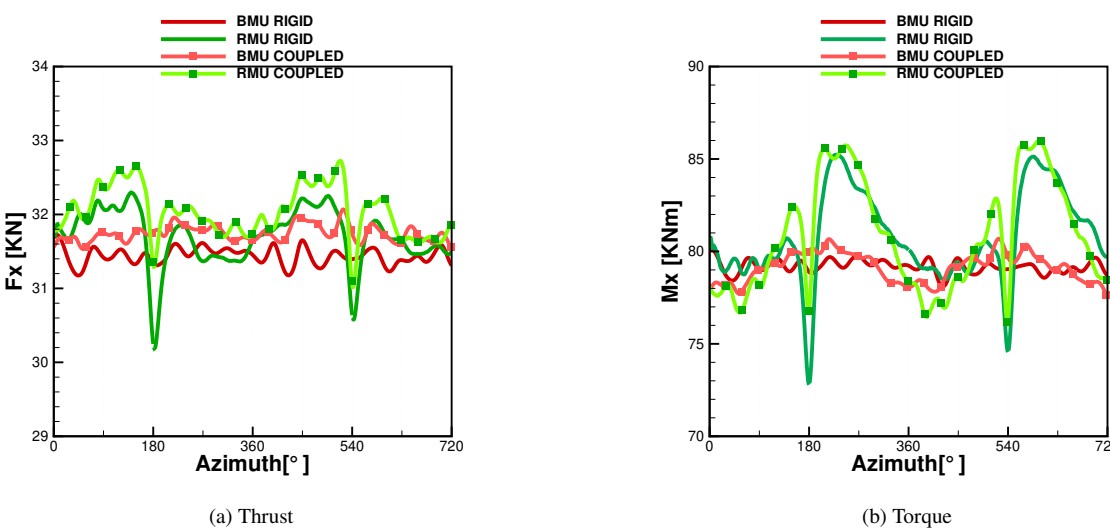

(a) Thrust

(b) Torque

**Figure 7.** Thrust and torque calculated with CFD at 6.1 $m/s$ in comparison BMU vs RMU, both rigid and coupled

The tip deformations in edge-wise direction are only dependent on the gravitational forces and show therefore almost no
difference between BMU and RMU. The same happens for the torsion, whose minimum value is slightly lower in RMU with
245 a really low maximum value of $0.075°$.

Regarding the global thrust and torque in the BMU case for rigid and coupled conditions, it can be seen that $M_x$ in fig.
7 has an oscillatory trend, directly related to the sinusoidal oscillation of the blade. The global thrust is slightly shifted to
higher values in case of coupling, where the mean value increases of 1%. This is due to the deformation of a pre-bended blade,
resulting in an increase of the effective rotor surface. Even if the torque oscillates more in the flexible case than in the rigid
state, the average difference is lower than 0.1% and therefore negligible. The RMU case shows a larger oscillation due to the
tower passage, and as in the BMU case, the structural coupling leads to a shift of both thrust and torque curves to higher values.
In particular, directly before the tower passage, the flexible blade reaches higher values of thrust (in average 1 to 2% more)
with a consequent higher thrust in front of the tower (in average 2 to 3% more). The same effect, although less evident, can
be seen for the torque. Averaging over three revolutions, the maximum difference in the produced power is up 2.3% and can
be seen between BMU R (rigid) and RMU C (coupled). Lastly, the difference in the sectional loads, averaged over the last
revolution, is analyzed in fig. 8. These are the sectional forces normal and tangential to the rotor plane.

The normal forces in coupled and rigid conditions show almost no difference. In the tangential loads, the one responsible
for the power at the shaft, a small increase (around 1%) can be observed between 40% and 60% of the blade radius, due to a

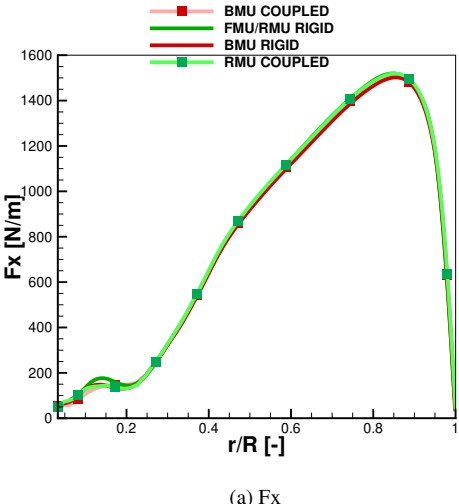

(a) Fx

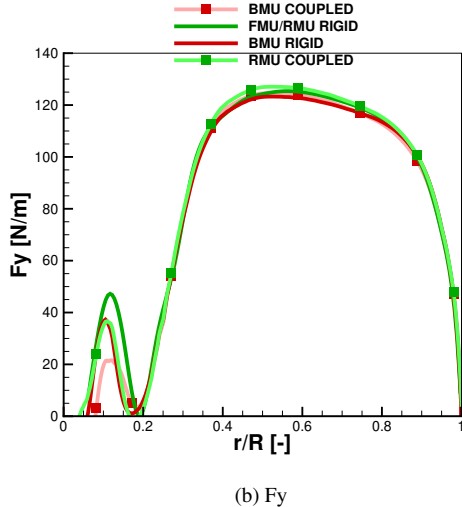

(b) Fy

**Figure 8.** Normal and tangential time averaged sectional loads calculated with CFD at 6.1 $m/s$ in comparison BMU vs RMU, both rigid and coupled

local slightly higher angle of attack (around 0.8 % more), connected with the positive value of torsion showed before, and due
to the increase of the effective rotor area.

While the CFD calculation have been made based on the operating conditions of the DANAERO experiment, further sim-
ulations have been conducted using BEM in order to determinate the generalization level of the results. Tip deformations in
flap-wise direction can be seen in fig. 9a, 9b and 9c. where an oscillation from 2.3% to 2.5% of the blade radius can be observed
as in CFD. In these BEM calculations the tilt angle needs to be in either both aerodynamic and structural models or in none of
them, therefore the only difference between BMU and RMU is the blade-tower passage effect. Differently from CFD, where
the impact was almost negligible, large oscillation occur due to the blade-tower passage, that already for the case with an inflow
velocity of 6.1 $m/s$ decreases up to 10 % (in comparison to no tower shadow). Increasing the inflow velocity and the RPM,
these oscillations become strong enough to preclude the deformations to reobtain the same shape as in BMU. An overestimated
blade-tower passage effect can be observed in the produced torque too, see fig. 9d, 9e and 9f. In particular, while with CFD
a reduction of this effect was observed when the structures were flexible (by low inflow velocity), this is not appearing using
BEM, that shows only an increase of it for high velocities of around 11 % (see fig. 9f). At the same time, while flexibility
shows almost no effect on the average torque at low velocities, up to 6% difference can be observed at 13 $m/s$. Especially in
this case it can be seen that the RMU C case converges back to the sinusoidal form of BMU C after a time equivalent to 150
degrees in which this oscillation is damped out.

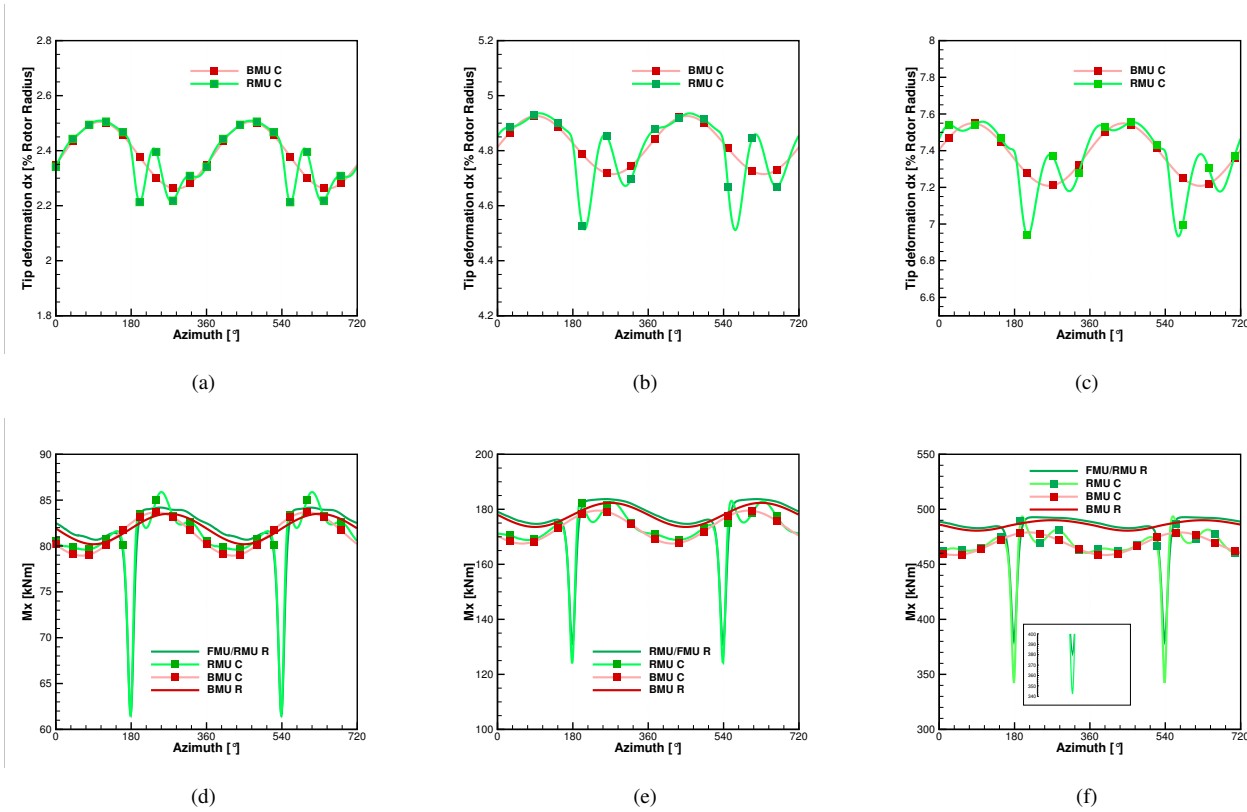

**Figure 9.** Aero-elastic calculations using BEM as aerodynamic model. Tip deformations in flap-wise direction BMU vs RMU: 6.1 $m/s$ in (a), 9.0 $m/s$ in (b) and 13 $m/s$ in(c). Torque ($M_x$) generated by one blade BMU vs RMU: 6.1 $m/s$ in (d), 9.0 $m/s$ in (e) and 13 $m/s$ in (f).

### 3.1.2 RMU vs FMU

As mentioned in section 2.5, the difference between RMU and FMU consists on the flexibility of tower and nacelle. The flap-wise, edge-wise and torsion deformations in comparison between RMU and FMU can be seen in fig. 10. Due to the low inflow velocity, the tower deflection contributes only 0.1% of the blade radius to the total blade out-of-plane deflection.

Considering the edge-wise deflection, the average value increases from 0.43% of the blade length for RMU to 0.65% for FMU due to the additional contribution of the tower top deformation. For the same aforementioned reasons, the torsion deflection has in average the same value, but due to the tower's torsion contribution, it shows a higher amplitude of the oscillation that increases in the FMU case up to 17% more. The global thrust ($F_x$) and torque ($M_x$) can be seen for the RMU and FMU rigid and coupled conditions in fig. 11, where almost no difference is shown between RMU and FMU coupled, due to the small deflections of the tower top.

As for the difference in FMU between rigid and coupled conditions, it can be seen that the decay due to the tower passage decreases by 6% (difference in $M_x$ between rigid and coupled at 180°). This has a direct effect on the maximum value reached

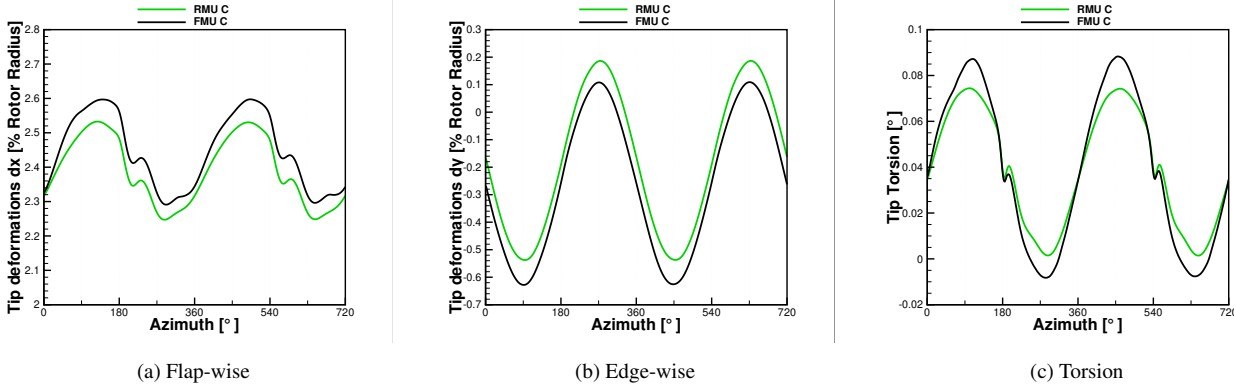

(a) Flap-wise        (b) Edge-wise        (c) Torsion

**Figure 10.** Tip deformations calculated with CFD at 6.1 $m/s$ in comparison RMU vs FMU

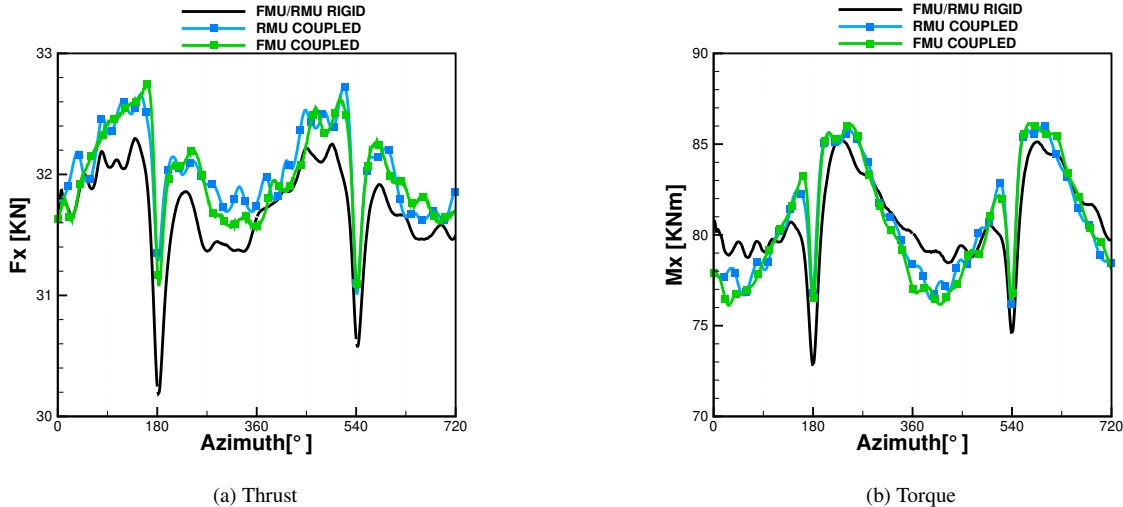

(a) Thrust             (b) Torque

**Figure 11.** Thrust and torque calculated with CFD at 6.1 $m/s$ in comparison RMU vs FMU, both rigid and coupled

directly after the recovery, which is also always higher than in the rigid case. It can also be observed that within one revolution the amplitude of the oscillation is higher in the coupled simulation. By averaging the results over the revolutions, it is found that the coupled case produces 3.5% more power than the rigid case. In order to understand this behavior, the averaged sectional

loads of the FMU rigid and coupled cases are compared, see fig. 12. The area of interest is from 20% of the blade radius, because near the hub the difference between the two curves is mostly due to the strong unsteadiness affecting the hub region, where separation is occurring. The loads in normal direction $F_x$ are not affected at all by the coupling. In contrast, the tangential loads $F_y$, the ones generating the torque $M_x$ and therefore the power, show some difference in the range between 40% and 70% of the blade radius (around 2 % more). This effect was also discussed by Sayed et al. (2016), who explained it with a slight

increase of the angle of attack in this region that is confirmed in pressure distributions at 40% and 50% of the blade radius in

fig. 13. A maximum $c_p$ difference of around 2.5% in the pressure side can be noticed. Considering that differently from Sayed et al. (2016), no decrease of the AOA is occuring in the outer region of the blade (for this inflow conditions), no compensation of this effect occurs and together to the increase of the rotor disk area, the increment in produced power is explained.

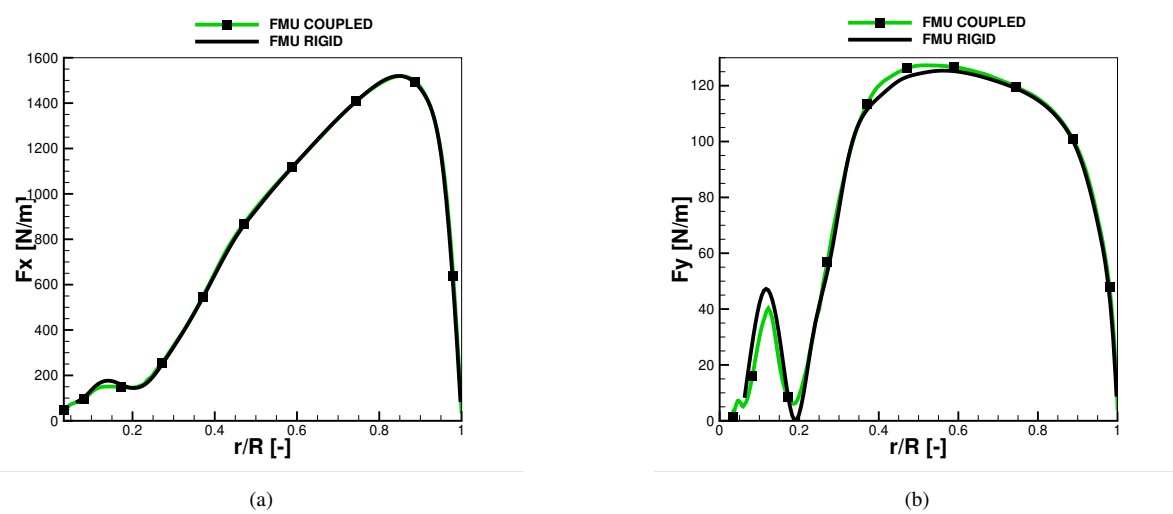

**Figure 12.** Sectional loads comparison in FMU both rigid and coupled calculated with CFD at 6.1 $m/s$.

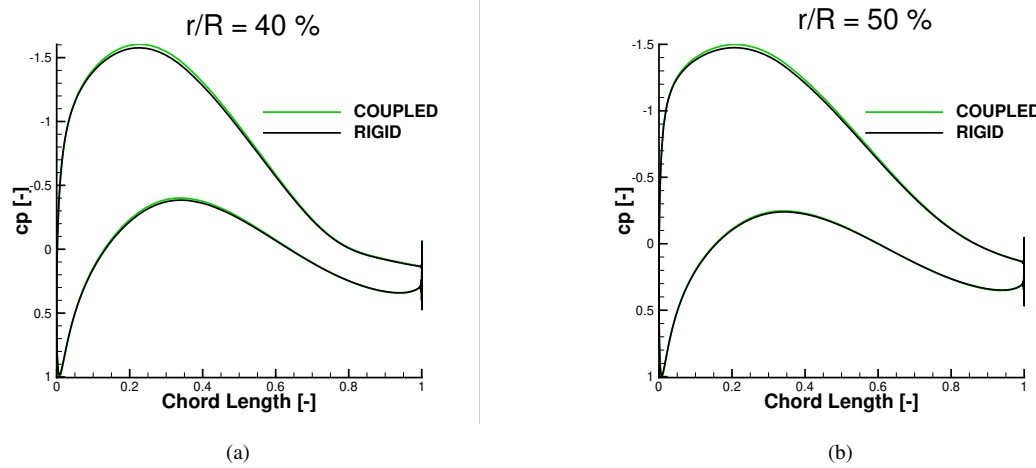

**Figure 13.** Pressure distributions for FMU rigid and coupled in comparison calculated with CFD at 6.1 $m/s$.

As in section 3.1.1, the simulations including the tower and its flexibility have been repeated using BEM and two more cases at higher inflow velocities have been added. As it can be seen in fig. 14a, 14b and 14c, almost no tower influence can be seen in the total blade deformation, because the predicted tower top deformation by AeroDyn is really low. Therefore, almost no difference can be noticed between FMU C and RMU C in the produced torque, but only the flexibility effect that increases

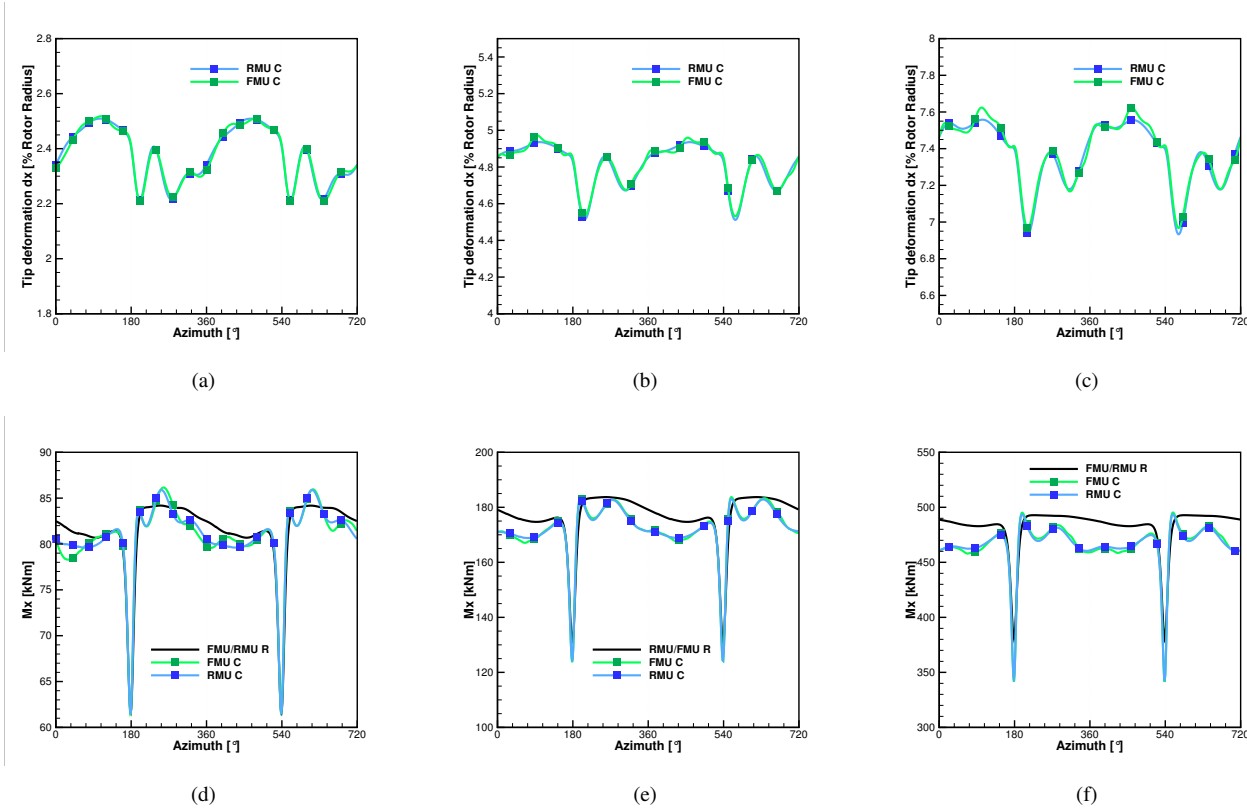

**Figure 14.** Aero-elastic calculations using BEM as aerodynamic model. Tip deformations in flap-wise direction RMU vs FMU: 6.1 $m/s$ in (a), 9.0 $m/s$ in (b) and 13 $m/s$ in(c). Torque ($M_x$) generated by one blade BMU vs RMU: 6.1 $m/s$ in (d), 9.0 $m/s$ in (e) and 13 $m/s$ in (f).

with the inflow velocity leading up to 6 % less power produced in comparison to rigid. Again, no decrease of the blade-tower passage effect can be noticed by 6.1 $m/s$, but only its increase at high velocity. Differently from CFD, the predicted torque
using BEM in the flexible case is always lower than the rigid case, and the curves show less oscillation than in CFD because of the lack of time-dependent 3D effects that BEM cannot capture.

### 3.1.3  FMU vs FMT

Figure 15 shows iso-surfaces of the $\lambda_2$-criterion for both inflow cases. The interaction can be seen between the near wake vortices and the Karman vortex street of the tower. The tower faces not only the turbulence of the flow, but also the wake
generated by the blades, resulting in a strongly turbulent flow and oscillations in the computed loads.

The comparison of the tip deformations in flapwise and edgewise directions and the torsion can be seen in fig. 16. The FMU case reaches a periodic steady state already after 2 revolutions, oscillating flap-wise with an average of 2.45% of the blade length. The same convergence trend can be seen for the edge-wise deformation and for the torsion, both of them almost negligible. All three are oscillating according to the rotational frequency.

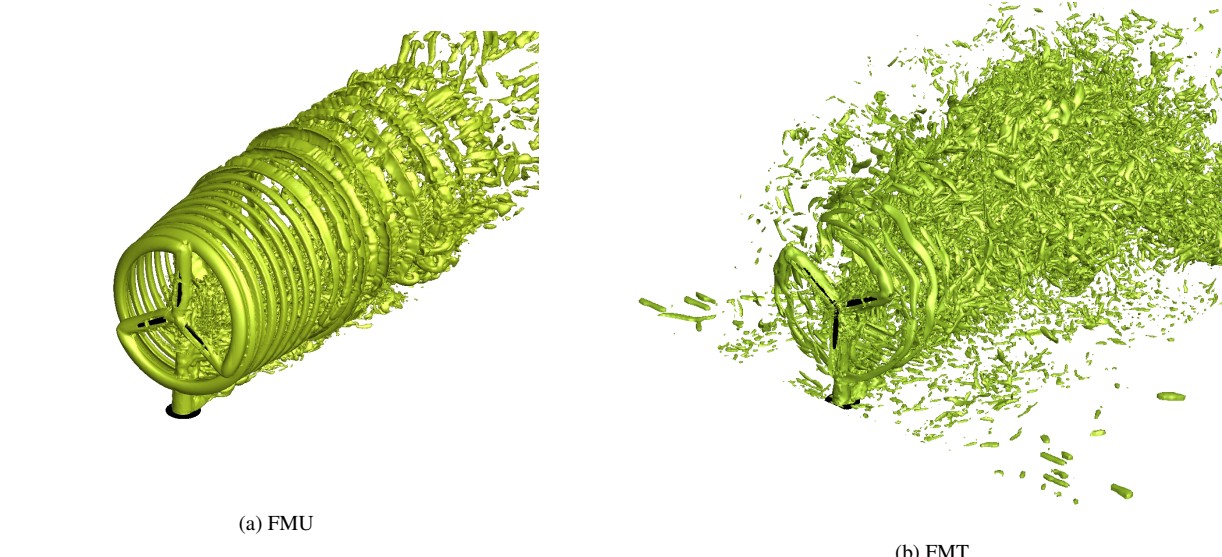

(a) FMU

(b) FMT

**Figure 15.** Visualization of the $\lambda_2$ criterion

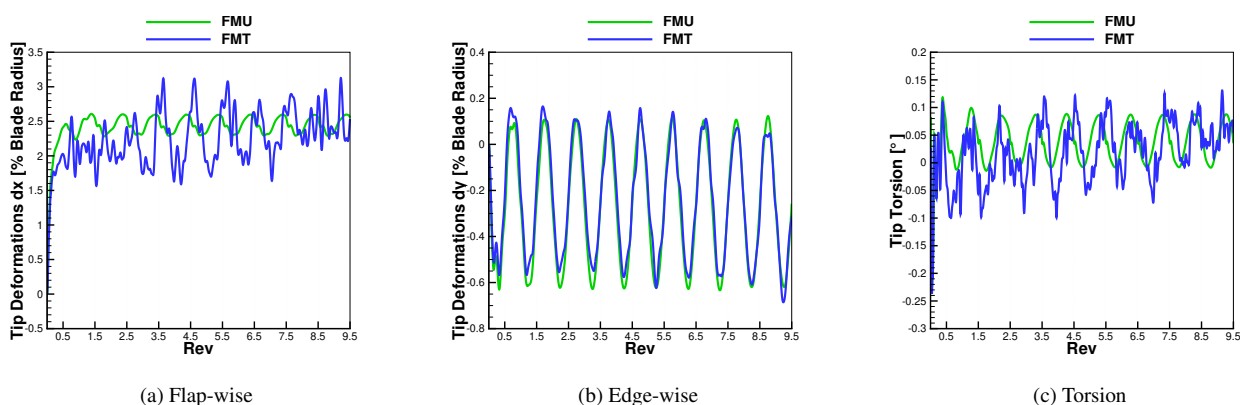

(a) Flap-wise

(b) Edge-wise

(c) Torsion

**Figure 16.** Tip deformations in comparison FMU vs FMT calculated with CFD.

The flap and torsion deformations are mostly affected by the presence of turbulence. Especially in the flap direction, 5 major peaks in 10 revolutions can be observed where the maximum deformation is around 3.1% of the blade length, that is 47% higher than the maximum in the uniform case. At the same time, the minimum flap-wise displacement, that is not due to the tower passage, is 30% lower than in the uniform case. For the torsion deformations, the turbulence is mostly affecting the minimum, that for FMU is -0.008°, while it is -0.09° for FMT. In the defined coordinate system, a negative torsion moves the

trailing edge more downwind. The edge-wise displacement, although in both cases oscillating around a mean value of 0.22%, has higher values for the first 8 minima of FMT.

This can be explained by the tower top deformations in flap-wise direction in fig. 17. In FMT the tower displacement is always smaller than in the FMU, and the tower deflection has an additional tilting effect on the rotor and consequently on the gravitational forces. After the eighth revolution, the tower top shows larger peaks in FMT than in FMU, leading to the opposite effect of a smaller peak in the edge-wise deformation.

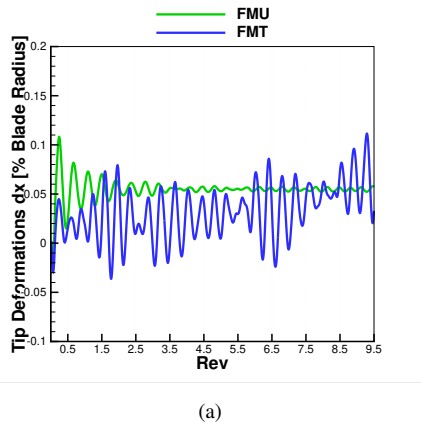

(a)

**Figure 17.** Tower top deformation in flap-wise direction calculated with CFD.

The spectra of the deformations is depicted in fig. 18, where the rotor frequency together with the higher harmonics are marked by a symbol. High amplitudes of the harmonics of the rotor frequency can be seen in flap-wise direction, where the first one is particularly strong. Additionally, it can be recognized that due to the inflow turbulence in FMT, the higher harmonics of the rotor frequency are obscured in the broadband of the spectrum. In edge-wise direction, that is mostly influenced by gravitation and not from aerodynamics, no strong increase can be seen for the rotor frequency, and the same happens for the torsion. On the other hand, the broadband has higher amplitudes in FMT than in FMU.

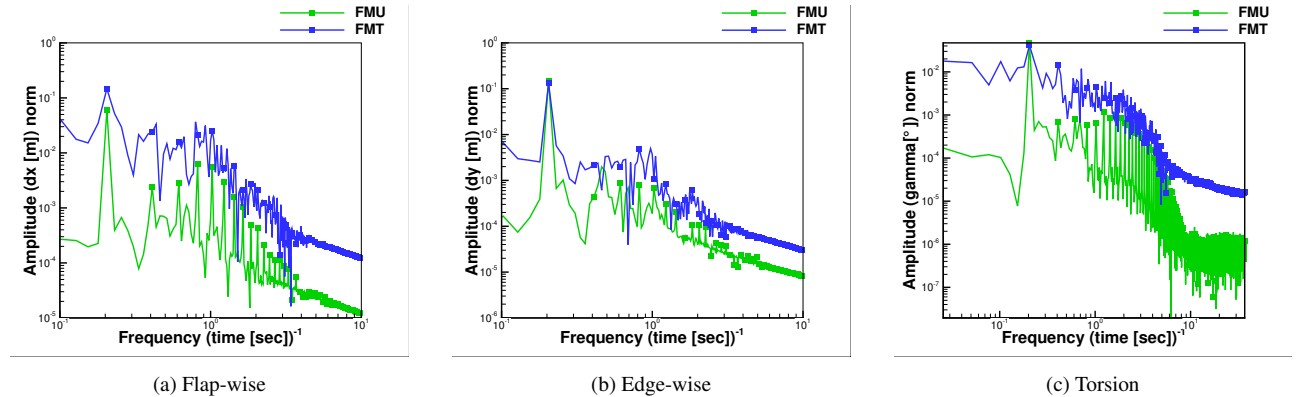

(a) Flap-wise      (b) Edge-wise      (c) Torsion

**Figure 18.** Spectra of the deformations in comparison FMU vs FMT

The effect of the tower can be again recognized in both FMU and FMT with a delay of around 20°, where a sudden drop in the tip deformations can be seen in fig. 16. Nevertheless this drop is almost negligible in comparison to the total affecting oscillation.

The loads resulting from the above described deformations of the FMT case are shown in fig. 19 (the FMU case has been already discussed in section 3.1.2). Independently of the rigidity of the structure, the turbulence leads to a much higher amplitude in the oscillation of the loads in comparison to FMU as seen in fig. 11. In fact, the torque $M_x$ fluctuates between 140 kNm and 10 kNm, while in FMU it ranges between 86 kNm and 72 kNm. Due to this high oscillation, the blade-tower passage can be hardly recognized. Unlike in the FMU case, the addition of flexibility has not marked consequences neither in thrust nor in

torque. Some peaks are increased in the flexible case, e.g. in both thrust and torque at 250°, 315°, 700° and 1000°. Averaging the result in time, the torque is increased by 2.5% (against 3.5% in the uniform case) due to flexibility. As for the blade-tower passage, the fluctuation inducted by the turbulence is the predominant source of oscillation; the flexibility represents only a secondary cause. This is valid only for the present case, where the inflow velocity and therefore the consequent deformations are small. In fig. 20, the sectional loads averaged over the same revolution for both rigid and coupled conditions are plotted. It

can be seen that although the shape of $F_y$ has changed between 30% and 70% of the blade length due to the strong oscillation brought by the turbulence, almost no difference is observed by the inclusion of flexibility in comparison to the uniform case as in fig. 12.

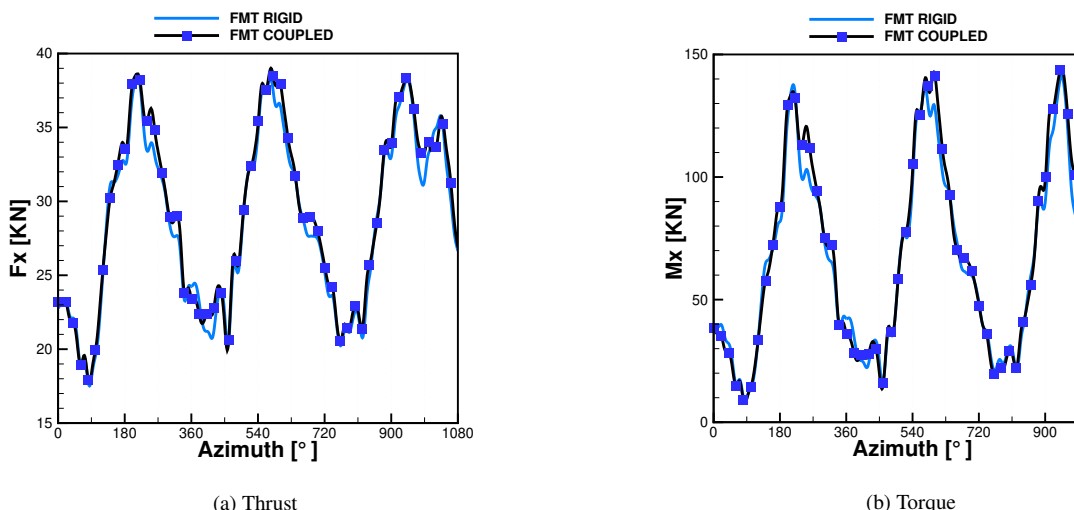

(a) Thrust                            (b) Torque

**Figure 19.** Global loads in FMT: comparison between rigid and coupled calculated with CFD.

## 3.2   DEL analysis

For the fatigue loading study of the different considered cases, the necessary constants described in section 2.6 have been set

to $Neq = 10^5$, $S_{m,eq} = 0$ and $m = 11$, where the last one is material dependent. The first two, as described in Hendrinks et al.

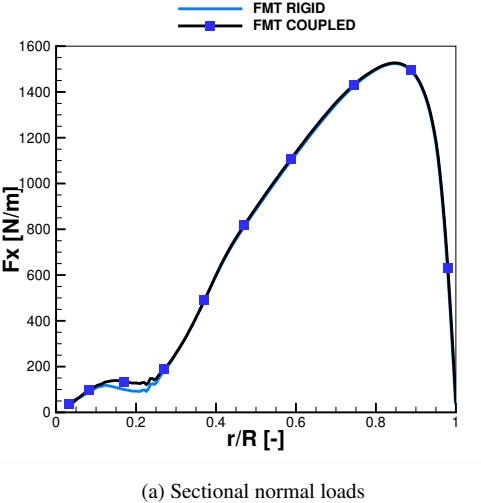

(a) Sectional normal loads

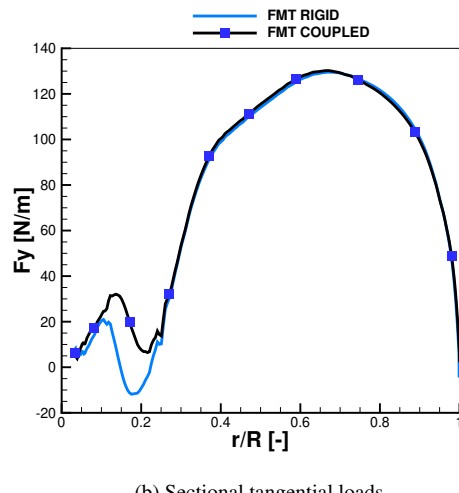

(b) Sectional tangential loads

**Figure 20.** Sectional loads in FMT: comparison between rigid and coupled calculated with CFD.

(1995), do not influence the results, because when making fatigue comparison, it is not the absolute value, but the ratio between the output from two signals, that is of interest. In order to consistently compare the cycle counts, the last three revolutions of each simulation case have been considered. The chosen input signals for the following analysis are the flap-wise and edge-wise blade root moment, $M_y$ and $M_x$ respectively. The first signal represents an unwanted action of the wind on the blade, while
the second one is responsible for the power production.

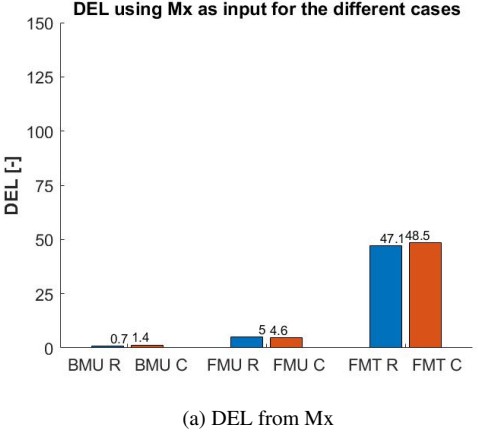

(a) DEL from Mx

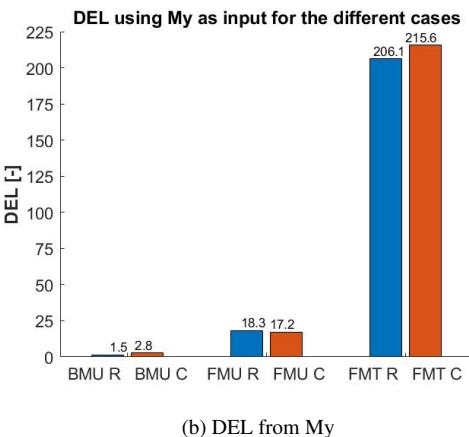

(b) DEL from My

**Figure 21.** DEL calculation based on CFD for the different cases using in (a) $M_x$ and in (b) $M_y$; $R$ and $C$ stays for "rigid" and "coupled"

The results are shown in fig. 21 and switching in BMU from rigid (R) to coupled (C), doubles the DEL, independently of the used input variable. It is observed in fig. 22a that the flexibility increases mainly the number of small cycles of the signal (fluctuations) and adds a few cycles with higher amplitude. In the case of FMU, already in rigid, DEL is increased by 7 times

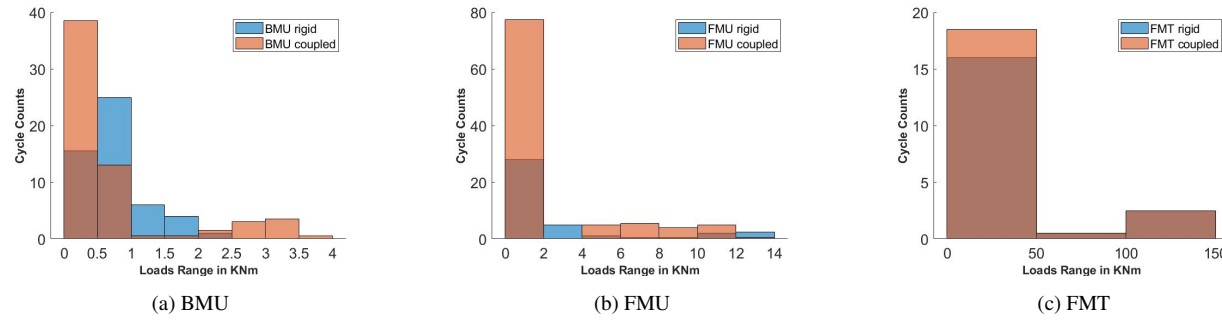

**Figure 22.** Comparison of number of cycle counts to load ranges using $M_x$ from CFD as input

in comparison to BMU, due to the tower passage and this effect is more pronounced using $M_y$ as input. It is interesting to
observe that in this case, the coupling has almost no effect on the total damaging. This is because, as shown in section 3.1, the
flexibility has two opposing influences on the loads: on the one side the increase of the oscillations and their mean value, and
on the other side the decrease of the blade-tower passage effect. These two effects almost counter act each other leading in total
to a comparable value of fatigue.

Switching the FMT case from rigid to flexible increases the DEL, because, as seen in fig. 22c, the flexibility adds a few more
small cycles but no big cycles, that are completely dominated by the impact of turbulence. Independently from the chosen
input, the addition of turbulence drastically increases the fatigue. Much fewer cycles are detected by the rainflow counting, but
they all have an amplitude larger than the largest cycles in FMU and BMU.

Finally, the ability of BEM of predicting the fatigue loading for the BMU and FMU cases is discussed. As it can be seen in
fig. 23a, BEM predicts slightly higher fatigue for BMU using $M_x$ as input signal than in CFD and that is because, as prescribed
in section 3.1.1, the BEM model presents a tilt angle also in the BMU case (differently from CFD), leading to a sinusoidal
oscillation of the forces. That means that altough the CFD calculations present many more smaller cycles due to unsteady
3D effects, the DEL is mostly affected by the big ones. The same impact, but more pronounced can be seen in BMU using
$M_y$ as input signal. This shows that modelling the turbine as a single blade in CFD when a tilt is given, can lead to a high
underevaluation of the fatigue.

Differently in the FMU case (no tilt modelling problem occurs), where for both rigid and coupled and for both chosen
input signals, BEM predicts higher fatigue than CFD. The difference between the rigid and coupled case remains the same as
predicted by CFD (so almost none), but the single values are almost two times the one from CFD. The reason for this can be
explained looking at the cycle count in fig. 23c. Although BEM predicts a smaller number of short cycles than CFD, cycles
with around 25 kNm appear, influencing mostly the fatigue calculation. Those cycles represent the blade-tower passage, which
effect shows to be overestimated by AeroDyn in comparison to CFD and therefore leads to higher DEL values.

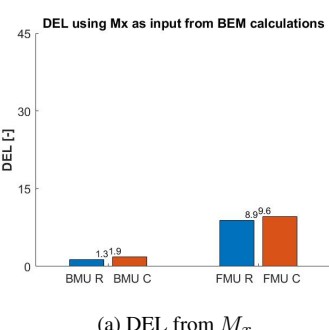
(a) DEL from $M_x$

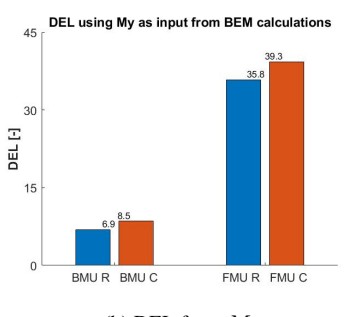
(b) DEL from $M_y$

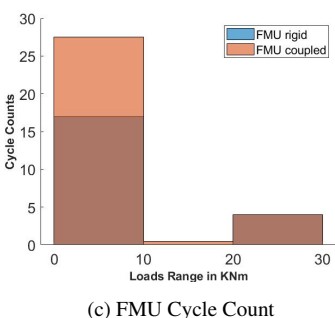
(c) FMU Cycle Count

**Figure 23.** DEL calculation using BEM: results for $M_x$ in (a) and for $M_y$ in (b). Cycle count in comparison to load ranges for FMU using as input $M_x$ in (c).

## 4 Conclusions

R1:G1 In the present work, different CFD models ranging from a single blade to the complete turbine including nacelle and tower of the DANERO turbine rotor were generated and coupled to a MBD structural model of the same turbine, by means of a loose (explicit) coupling. The aeroelastic response of the reference turbine was calculated by the use of models increasing their complexity and fidelity in order to recognize differences and deviations connected to modelling approaches which computational and pre-processing costs strongly differ. The effects of turbulent inflow conditions were analyzed in comparison to uniform inflow, considering both a rigid and a completely elastic wind turbine model. Additionally, a BEM model of the turbine was consistently generated and assessed against the CFD results. In this way it was possible to consider additional uniform inflow cases to determinate the generalization level of the results. The objective of this study was to identify the impact and interaction of the different components and modelling approaches on the transient loads and on the DEL of the only blade. This was evaluated taking into account the flap-wise and edge-wise blade root moment at the rotor center. The major results of this study can be summarized in the following:

1. A high-fidelity FSI model of the DANAERO wind turbine has been generated and validated in comparison to experimental results.

2. Modelling the turbine as a single blade instead of entirely leads to only around $1\%$ to $2\%$ difference in the average quantities (sectional loads, average torque and deformations). Differently, the resulting DEL increases from BMU to RMU up to 12 times due to the additional large cycles induced by the tower passage and because of the consideration of the tilt angle that leads to a sinusoidal oscillation of the loads, as showed by the BEM calculations.

3. The introduction of flexibility in BMU increases the DEL because of more loads oscillations, that in FMU are balanced by a reduction of the tower effect. That is why the DEL showed not to be affected by flexibility in this case.

4. When the entire turbine is computed as flexible, a slight increase of the torque is found in comparison to the rigid case at the computed low inflow velocity, due to the increase of the rotor disk area and a slightly increase of the AOA.

5. BEM shows in general a good agreement with CFD in evaluating the average quantities, although an overestimated tower effect is predicted (with the standard tower model implemented in the AeroDyn version coupled to SIMPACK) with direct result on the DEL evaluation. Additionally, CFD shows a decrease of the tower effect with the introduction of flexibility, that BEM is not showing.

6. Comparing uniform and turbulent inflow, the spectra of the blade tip deformations show that the turbulence is increasing the amplitude of the broadband, while obscuring the higher harmonics of the rotor frequency.

7. Independently of the rigidity of the turbine, turbulence leads to a much higher amplitude in the load oscillations, in which the tower passage becomes only a neglectible effect. This has a direct result on the DEL of the blade that increases up to 11 times in comparison to FMU. Flexibility is indeed additionally increasing the fatigue, but much less in comparison to what turbulence does, showing that this is the main factor influencing the DEL calculation.

R1:G1a In general it can be concluded that, in the computed cases, turbulence showed to be the most important factor influencing the DEL of the single blade, more than flexibility that played in comparison only a marginal role for this specific case where the rotor radius is only 40 m long. Note that when the rotor size increases, the effect of flexibility may play a greater role. Also, the modelling of the turbine as a single blade strongly underestimates the DEL, even if CFD is used. On the other side, a single blade model (that is much cheaper than a full CFD model of the turbine) realizes to give valid results when just the averaged deformations and loads in uniform inflow are of interest and the predicted tower top deformations are low (as for the low inflow velocity studied in this paper). AeroDyn overestimates the blade-tower effect in comparison to CFD, leading to higher fatigue values, but excluding this overestimated tower effect, BEM realizes to give useful conclusions regarding the effect of flexibility on fatigue for the uniform inflow conditions at which it has been used.

*Author contributions.* G. Guma generated a part of the CFD model, the MBD model, ran the coupled simulations, and performed the post processing and analysis. G. Bangga generated a part of the CFD model. T. Lutz and E. Krämer supported the research, defined and supervised the work and revised the manuscript.

*Competing interests.* The authors declare that they have no conflict of interest.

*Acknowledgements.* The authors gratefully acknowledge the DANAERO Consortium for providing the geometry and structural data. They acknowledge additionally SIMPACK for providing the user licenses and the funders of the project WINSENT (Code number: 0324129), the Federal Ministry for Economic Affairs and Energy (BMWi) and the Ministry of the Environment, Climate Protection and the Energy Sector Baden-Württemberg under the funding number "L75 16012", under which project improvements on the simulation chain were performed. Computer resources were provided by the Gauss Centre for Supercomputing/Leibniz Supercomputing Centre under grant "pr94va". Additionally a particular thank is given to the DLR and SWE-University of Stuttgart for the productive discussions that helped improving

the structural model of these simulations. Finally the authors would like to acknowledge Aimable Uwumukiza for his effort in correcting the language and exposure of this paper.

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
