# Peer review of "Aero-elastic analysis of wind turbines under turbulent inflow conditions"

_Wind Energy Science, 2020_

## Referee Comment (RC1) · Anonymous Referee #1 · 2 Apr 2020

Interesting paper discussing the effect of flexibility and inflow turbulence on aero-elastic behavior. Impressive simulations with CFD coupled to a MBD code.

General comments

-English langauge (especially sentence constructions) to be checked by native speaker

-The impact of different CFD modeling (BMU vs RMU) on results is very useful. However can some of the main conclusions on the influence of flexibiliy and turbulence also be achieved with lower fidelity aerodynamic models like BEM (i.e. is CFD really needed to arrive at the conclusions given)? And to what extent are these conclusions trivial rather than new insights? As the results are now based on only 1 operational condition, can these be generalized? Perhaps BEM simulations can aid to obtain results for a larger operational regime?

Specific comments

-p5 2.2.1 Line 111 Not clear what SIMBEAM means

-p5 2.3.1 Fig. 2 This seems more a high level visualization of the model rather than showing details

-p6 2.4 Table 1 Perhaps good to signify coupled and rigid in this table as well? E.g. although FMU is said to have flexible components later on in the paper also FMU-rigid pops up.

-89 3.1.1 Line 172 Not sure if I understand. Observing Fig. 5a shows a minimum at approx 280deg which is more than 20deg past 180deg??

-89 3.1.1 Line 179 This explanation would be placed better somewhere at the start of this section when fig. 5 is firstly introduced.

-p9 3.1.1 Fig 5 If we focus on the effect of different aerodynamic modeling on temporal variations, why not also show time variation of aerodynamic variables (e.g. blade root moments integrated from pressures) rather than only deformations?

-p9 3.1.1 Fig 5 Why not show a 360deg range to focus better? (also holds for several other figures)

-p9 3.1.1 Line 187 Add 'Figure' between 'in' and '6'

-p9 3.1.1 Fig 7 Indicate in caption we are looking at time averaged forces

-p10 3.1.2 Line 207 Not really clear what text aforementioned relates to

-p11 3.1.2 Line 213 Is the reason for the 6% decrease in decay clarified (what is meant exactly with decay here, would deformation not lead to a smaller distance to the tower and larger tower effect)?

-p15 3.1 Line 258 Add 'Figure' between 'see' and '9'. Is the refernce to figure 9 correct?

-p15 3.1 Line 266/267 Not sure if I observe an Fy shape change between 30%-70% in Figure 16?

-p17 4 Line 294 Probably 'larger' instead of 'major' is meant (also in abstract). It is not clear from the sentence which effect is meant exactly (fatigue loads of My?).

---

## Referee Comment (RC2) · David Verelst (Referee) · 7 Apr 2020

Please note that my expertise area is within hydro-servo-aero-elastic load computations and as such I can't comment on the CFD methodology.

Additional comments can be found in the attached PDF as notes. This abstracts needs additional attention regarding language and grammar, I have marked some grammatical incorrect sentences in the attached PDF.

**General**

If I understood your work correctly, you have performed an FSI study with various modelling approaches, from a "simple" blade (no turbine present) to the full complexity of the turbine (with tower, shear, turbulent inflow). You have used a model of the

DANAERO turbine. You conclude that the considered model fidelities affect the loading and consequently fatigue computations.

Why have you conducted this study with CFD, and not just simple BEM? Would you expect the conclusions you have drawn to be significantly different? You could consider adding BEM computations to many of the cases you have considered, and I think this could add something of value to the paper.

Please consider finding a better balance between literately describing what happens in the plots, and adding more discussions regarding what is physically happening in the different modelling scenarios.

**2.3.1 Structural model**

Did you verify the structural eigenfrequencies and damping match the experimental setup of the DANAERO turbine? Will be relevant for the fatigue response.

**3.1 Aeroelastic effects**

Could you indicate more explicitly what the aim of this section is? I assume its purpose is to validate the model?

Can you elaborate on the challenges when comparing turbulent measurements with simulations using deterministic inflow?

**3.2 DEL Analysis**

It is not clear to me how you have performed the analysis:

- According to section 2.4 on the first paragraph of page 6, you computed 6 revolutions for BMU and 10 for the other cases (FMU/FMT). That means you have a different simulation length, how did you account for that?

- Why are using a reference number of cycles of Neq=1e5 while your time series are that short? Wouldn't it make more sense to compute a 1Hz equivalent load?

- In Figure 18 you clearly show that the binning of the cycle counts is very different. Can you comment on that?

Please also note the supplement to this comment:
https://www.wind-energ-sci-discuss.net/wes-2020-22/wes-2020-22-RC2-supplement.pdf

**Supplement:**

[revised manuscript text omitted]

---

## Referee Comment (RC3) · Anonymous Referee #3 · 8 Apr 2020

**Review of paper wes-2020-22:**

**Aero-elastic analysis of wind turbines under turbulent inflow conditions**

by Giorgia Guma[1], Galih Bangga[1], Thorsten Lutz[1], and Edwald Krämer[1]

[1]Institute of Aerodynamics and Gas Dynamics, University of Stuttgart, Pfaffenwaldring 21, 70569 Stuttgart

**Brief summary**

The author's present a high fidelity FSI model based on the CFD code FLOWer and the structural multibody code SIMPACK in an explicit coupling. Different complexities of the FSI model are presented from a blade alone model over rotor model to full modelling of the rotor, nacelle and tower. Uniform inflow as well as turbulent inflow cases are simulated. The simulations are carried on the so-called DANAERO rotor, a 2MW NM80 turbine with an 80 m rotor.
Comparing the different simulated cases, the impact on Damage Equivalent Loading (DEL) of turbine flexibility and turbulence is discussed. For the relative low wind speed case simulated with turbulence it is concluded that turbulence has, by far the highest impact on the DEL compared with impact of flexibility.

**Overall comments**

The overall subject of presenting a high fidelity FSI model is of considerable relevance for the research community as the turbines are continuously upscaled leading to more flexible designs where the aeroelastic effects become more and more important and require such FSI models to be truly analysed. The authors also mention this motivation in the introduction.

However, a limitation of the paper content is that the simulations are carried out on a turbine design that is more than 20 years old and with a structural design that is much stiffer than designs that are more recent. Also the size of 2MW and a diameter of 80m for the chosen turbine is quite different from both recent reference turbine designs (e.g. the INNWIND and AVATAR reference rotors defined several years ago) in the research community as well as industrial designs that now have exceeded 10MW and 200m in diameter.

The actual choice of the DANAERO turbine might be because detailed aerodynamic measurements are available for this turbine but the experimental data are only used in one case in the paper.

So due to fact that the simulations are carried out on a relative stiff turbine design, some of the results and discussions of e.g. impact of flexibility on elastic torsion which are in the range of tenths of a degree, become somewhat theoretical without any real impact.

**Specific comments**

**Title**

The present title indicates that the paper has a considerable focus and part on simulating turbulent inflow conditions. However, it is a limited part of the paper so it could be considered to change the title, e.g. to:

**A High Fidelity Simulation Tool for Aeroelastic Simulations of Wind Turbine**

**Abstract**

Introducing the turbine used for the simulations as ".. the DANAERO turbine …" is not precise as many in the research community do not know that specific turbine.

So it should be changed to a real description of the turbine, e.g. "a 2MW NM80 turbine with an 80m diameter rotor".

Also the text: " .. specific case of the DANAERO experiment.." could be more informative, e.g. like:

" .. of the inflow turbulence is analysed for a specific case in comparison with experimental data from the DANAERO experiment."

**1. Introduction**

Satisfactory introduction and description of previous work within the field and the intro to the contents of the present paper at the end.

Minor typos

Line 22:

- .. where the only blades ..

Line 29:

- .. and the compared to BEM ..

**2. Methodology**

Section 2.2

Line 96-97

- The description of the turbulence generation by body forces is very brief – please expand, e.g. over what axial distance are the body forces applied and how much does the turbulence decay down to the turbine ?

Line 128

- What is the actual FMT time simulation length and what did determine the length ?

**3 Results**

3.1 Aeroelastic effects

Line 165-167

- The discussion of this case is very short – please expand
- Why isn´t this case simulated with turbulence as the experimental data are shown, probably from several revolutions in turbulent inflow

Line 173-174

- The percentage values are shown with two decimals. They are probably based on the time trace values so maybe better just to show one decimal or how accurate are these values ?

Line 187

- that Mx in 6 – please correct

Line 215

- An increase in power for coupled case of 3.5% is quite much – is it a time varying flow due to separation in the region of 40-50% radius or what can be the causes?

Line 248

- " .. in FMT a peak appears close to the first flap bending eigenfrequency of the blade (f1 = 0.938Hz) that could lead to instabilities .. ". Please show the position of that frequence by a vertical line in the spectrum.
- As concerns instability – how can this lead to instability ?

Line 282-285

- Could numerical stabilities in the coupling generate the considerable number of small oscillations?

**Final conclusion of review**

The reviewer can recommend publication of the paper considering to include the above comments/questions.

---

## Author Comment (AC1) · 10 Jun 2020

**Reply to comments by Reviewer Nr. 1**

Giorgia Guma on behalf of the authors
IAG, University of Stuttgart

June 10, 2020

The authors would like to thank the reviewer for his/her efforts and valuable comments. They are very much appreciated and incorporated into the revised paper.

In the present document the comments given by the 1st reviewer are addressed consecutively. The following formatting is chosen:

- The reviewer comments are marked in blue and italic.

- The reply by the authors is in black color

- A marked-up manuscript is added. Changed section with regard to the comments by reviewer 1 are marked in yellow. Changed sections with regard to comments by more reviewers are marked in gray.

**General comments**

1. *"English langauge (especially sentence constructions) to be checked by native speaker"*

The entire manuscript has been revised after your suggestion. For sake of clarity, the modifications have not been underlined with a color.

2. *"-The impact of different CFD modeling (BMU vs RMU) on results is very useful. However can some of the main conclusions on the influence of flexibiliy and turbulence also be achieved with lower fidelity aerodynamic models like BEM (i.e. is CFD really needed to arrive at the conclusions given)? And to what extent are these conclusions trivial rather than new insights? As the results are now based on only 1 operational condition, can these be generalized? Perhaps BEM simulations can aid to obtain results for a larger operational regime?"*

Thank you for your comment. It was decided to use CFD instead of BEM based on the results from the VortexLoads project, see [Boorsma et al.(2019)] and [Boorsma et al.(2020)]. Here it was shown that BEM overpredicts fatigue loading by rotors with a high induction. We agreed anyhow with you that it would have been interesting to add BEM calculations and therefore we now created an AeroDyn model of the turbine. This software was chosen because already coupled to Simpack. Please see $\boxed{\text{R1:G1}}$ (page 6, line 132) for the model descritption and $\boxed{\text{R1:G2}}$ (page 13, line 250), $\boxed{\text{R1:G3}}$ (page 16, line 288) and $\boxed{\text{R1:G4}}$ (page 22, line 357) for the results.

**Specific comments**

1. "*p5 2.2.1 Line 111 Not clear what SIMBEAM means*" Additional explanations according to the SIMPACK manual has been added at $\boxed{\textbf{R1:S1}}$ (page 5, line 119).

2. "*p5 2.3.1 Fig. 2 This seems more a high level visualization of the model rather than showing details*" Thank you we changed the caption in fig. 2.

3. "*p6 2.4 Table 1 Perhaps good to signify coupled and rigid in this table as well? E.g. although FMU is said to have flexible components later on in the paper also FMU-rigid pops up.*" We agree and changed the description of the table in $\boxed{\textbf{R1:S3}}$ (page 9, line 173).

4. "*p9 3.1.1 Line 172 Not sure if I understand. Observing Fig. 5a shows a minimum at approx 280deg which is more than 20deg past 180deg??*" We are referring to the local minimum at 2.3% of the blade radius. But you are right, writing just minimum is confusing, therefore we changed the sentence in $\boxed{\textbf{R1:S4}}$ (page 11, line 220).

5. "*p9 3.1.1 Line 179 This explanation would be placed better somewhere at the start of this section when fig. 5 is firstly introduced*" We agree and moved it to $\boxed{\textbf{R1:S5}}$ (page 11, line 222).

6. "*p9 3.1.1 Fig 5 If we focus on the effect of different aerodynamic modeling on temporal variations, why not also show time variation of aerodynamic variables (e.g. blade root moments integrated from pressures) rather than only deformations?*" Thank you for your comment. Every time we compared different aerodynamic models within another, we tried to go step by step consistently starting from the deformations, then the global Thrust and Torque time series, and at the end the averaged sectional loads over the blade. Keeping this structure, in our opinion, allows the reader to better focus on the different effects.

7. "*p9 3.1.1 Fig 5 Why not show a 360deg range to focus better? (also holds for several other figures)*" Thank you. In the FMT case for example, considering that the inflow velocitiy is changing, we think it is better to show a wider range in order to see its influence. We agree that in BMU, RMU and FMU, being subjected to uniform inflow, a steady state is reached that is repeating periodically. It was important for us to show that some behaviors in those cases where not related to a not completely converged solution, but really periodically repeating. We made a compromise and reduced the range to 720 deg (not for FMT).

8. "*p9 3.1.1 Line 187 Add 'Figure' between 'in' and '6'*" Thank you, we changed it in $\boxed{\textbf{R1:S8}}$ (page 12, line 235).

9. "*p9 3.1.1 Fig 7 Indicate in caption we are looking at time averaged forces*" Thank you, we added in the caption of fig. 8.

10. "*p10 3.1.2 Line 207 Not really clear what text aforementioned relates to*" We meant that the tower flexibility increases the amplitude of the total torsion oscillation of the blade. In order to make it clearer, the sentence has been revised in $\boxed{\textbf{R1:S10}}$ (page 14, line 270).

11. "*p11 3.1.2 Line 213 Is the reason for the 6% decrease in decay clarified (what is meant exactly with decay here, would deformation not lead to a smaller distance to the tower and larger tower effect)?*" The sentence was reformulated in $\boxed{\textbf{R1:S11}}$ (page 14, line 275) in order to explain what is meant in this case by decay. As you said, the deformation leads to a larger tower effect, that affects mostly the outer region. Nevertheless in our computations resulted that the flexibility has also the effect to increase the loads in the inner region, and the sum of the two effects is an overall increase of the loads distribution in time.

12. *"p15 3.1 Line 258 Add 'Figure' between 'see' and '9'. Is the refernce to figure 9 correct?"* Thank you, we changed it in $\boxed{\textbf{R1:S12}}$ (page 20, line 326). Yes the reference is correct, it refers to the loads in FMU showing that the absolute values and the oscillation amplitude is lower than in FMT, independently of the rigidity.

13. *"p15 3.1 Line 266/267 Not sure if I observe an Fy shape change between 30%-70% in Figure 16?"* It was meant in comparison to FMU, therefore fig 12. The text in $\boxed{\textbf{R1:S13}}$ (page 20, line 335) has been changed, in order to make it easier to understand it.

14. *"p17 4 Line 294 Probably 'larger' instead of 'major' is meant (also in abstract). It is not clear from the sentence which effect is meant exactly (fatigue loads of My?)."* We agreed and changed the sentence in $\boxed{\textbf{R1:S14}}$ (page 23, line 374). And yes, as you said this conclusion has been drawn from the fatigue loads of both My and Mx.

**References**

[revised manuscript text omitted]

---

## Author Comment (AC2) · 10 Jun 2020

**Reply to comments by Reviewer Nr. 2**

Giorgia Guma on behalf of the authors
IAG, University of Stuttgart

June 10, 2020

The authors would like to thank the reviewer for his efforts and valuable comments. They are very much appreciated and incorporated into the revised paper.

In the present document the comments given by the 2nd reviewer are addressed consecutively. The following formatting is chosen:

- The reviewer comments are marked in blue and italic.

- The reply by the authors is in black color

- A marked-up manuscript is added. Changed section with regard to the comments by reviewer 2 are marked in orange. Changed sections with regard to comments by more reviewers are marked in gray.

**General comments**

1. "*This abstracts needs additional attention regarding language and grammar, I have marked some grammatical incorrect sentences in the attached PDF.*"

Thank you very much for helping correcting it. The entire manuscript has been now revised, but for sake of clarity, the modifications have not been underlined.

2. "*Why have you conducted this study with CFD, and not just simple BEM? Would you expect the conclusions you have drawn to be significantly different? You could consider adding BEM computations to many of the cases you have considered, and I think this could add something of value to the paper.*"

Thank you for your comment. It was decided to use CFD instead of BEM based on the results from the VortexLoads project, see [Boorsma et al.(2019)] and [Boorsma et al.(2020)]. Here it was shown that BEM overpredicts fatigue loading by rotors with a high induction. We agreed anyhow with you that it would have been interesting to add BEM calculations and therefore we now created an AeroDyn model of the turbine. This software was chosen because already coupled to Simpack. Please see R2:G1 (page 6, line 132) for the model descritption and R2:G2 (page 13, line 250), R2:G3 (page 16, line 288) and R2:G4 (page 22, line 357) for the results.

3. "*Please consider finding a better balance between literately describing what happens in the plots, and adding more discussions regarding what is physically happening in the different modelling scenarios.*" Thank you for your comment. We tried to satisfy all your and other reviewers requests, but due to firstly "maintenance" problems and then huge security issues that involved

many supercomputers in all the world since weeks, it was not possible to add more physical discussions, because no access to the data was allowed. We do appreciate your interest and constructive comment and we will take it seriously into consideration for our next work.

**Specific comments**

1. "*Did you verify the structural eigenfrequencies and damping match the experimental setup of the DANAERO turbine? Will be relevant for the fatigue response.*" Thank you. After asking the permission to the project partners to show the measured eigenfrequencies, a table was added in the paper in $\boxed{\text{R2:S1}}$ (page 5, line 126). These have been compared also to the eigenfrequencies computed by other project partners using beam elements too, but with other softwares. Unfortunately, we can only show the comparison to the measured ones.

When non-linear SIMBEAM elements are used in Simpack, it is not possible to apply the damping factors directly to the modes, but either Rayleigh or Kevin-Voigt coefficients need to be calculated. The first method is the widely mostly used, and therefore it was chosen and $\alpha$ and $\beta$ have been calculated from the first and second eigenfrequency of the blade.

2. "*Could you indicate more explicitly what the aim of this section is? I assume its purpose is to validate the model?*" This section is an introductory section to the different cases that have been computed and compared. As you suggested, we used it also to show some comparison to the experimental results. This part has been extended with a larger explanation as you can see in $\boxed{\text{R3:S6}}$ (page 11, line 208)

3. "*Can you elaborate on the challenges when comparing turbulent measurements with simulations using deterministic inflow?*" We used a stochastic model based on Mann. That means that every time we create a Mann box, although the input parameters are the same, we get a different stochastic turbulence based on the same mean velocity. The description of the turbulence generation has been extended in $\boxed{\text{R3:S4}}$ (page 4, line 98). That means that simulations and experiments are not directly comparable because the time series of the inflow velocity are not the same. Therefore we compared the averaged results from the experiments to the averaged results from the simulations, as described now in $\boxed{\text{R2:S2}}$ (page 11, line 208).

4. "*According to section 2.4 on the first paragraph of page 6, you computed 6 revolutions for BMU and 10 for the other cases (FMU/FMT). That means you have a different simulation length, how did you account for that?*" The BMU case, being just a one blade model of the turbine, reached a periodicity and convergence of the loads and deformations much faster than in the FMU/FMTcases, with less than 1% difference in average between two revolutions. For these other two cases, more revolutions were necessary and that is why they have been computed longer. The DEL calculation was then made taking into consideration the last three revolutions for each case, in order to take into account the same simulation length. We added this in $\boxed{\text{R2:S4}}$ (page 21, line 341).

5. "*Why are using a reference number of cycles of Neq=1e5 while your time series are that short? Wouldn't it make more sense to compute a 1Hz equivalent load?*" This Neq is just a number, as long as it is kept constant between the different DEL calculations, the comparison between different signal inputs remains the same. Modifying it would just change the absolute value, that is anyhow not of interest.

6. "*In Figure 18 you clearly show that the binning of the cycle counts is very different. Can you comment on that?*" The rainflow counting was performed using a standard function available in MATLAB that does not allow to change the size of the edges. But we agree that forcing

the same size, would make the results and their interpretation clearer. Therefore we forced the output to have the same binning for clearness and changed pictures 22. Please consider FMT having the same binning size from the beginning is just random and not particularly of interest.

7. "*Line 108: I am not familiar with the modeling details of SIMPACK. Does the formulation account for the non-linear geometrical response of the blade? In HAWC2 the deformations within a body are considered linear, but the non-linear geometrical response is captured by using multiple bodies for a given body. Could you indicate how SIMPACK takes care of this? It could be this does not matter too much since I think the DANAERO turbine is not that flexible and the deflections are generally small (if I am not mistaken).*" Yes it is as you said, in principle it would make almost no difference to use a linear or non-linear model, because the turbine is really stiff. Anyhow for correctness we used "Non-linear" SIMBEAM elements. This is a new feature of SIMPACK, because until a few years ago, it was necessary to build the strucutral model exactly as you mentioned for HAWC2 in order to account for the non-linearities. Now the switch from linear to non-linear is much more intuitive. Some more details about it have been added in R1:S1 (page 5, line 119).

8. "*Line 5: This is somewhat confusing: I can not find any specific comparison with the DANAERO experiment other than a simple qualitative validation in figure 4*" The sentence has been a bit reformulated and additional comments have been added in section "Aeroelastic effects" R2:S6 (page 1, line 6)

9. "*Line 120: This could mean that the turbine will operating at unrealistic operating points at time, depending on how much pitch/rpm variations have been observed in the corresponding measurements. Could you maybe brefly comment on that?*" Yes that is completely true. The pitch and RPM are set at the inflow velocity of 6.1 m/s, that is also the inflow velocity at which the Mann box is generated. Although the TI of 20%, the inflow velocity is really low (6.1 m/s) and far away from cut-off. Therefore the controller would mainly change the RPM and not the pitch angle. The change in RPM has an influence on the full system natural frequencies, on the blade-tower passage frequency, and on the Thrust, that would increase with the RPM and therefore the flapwise tip deformations. This observation has been added in R2:S7 (page 8, line 152).

10. "*Line 113: What does it mean for the structural damping, do you get a similar result (expressed in log decr for example) as the given reference wind turbine?*" Please, see answer to the first comment.

11. "*Line 165: to help the reader, indicate which symbols you are using: normal $F_n$ and tangential $F_t$*" Thank you for the suggestion, it has been done, see R2:S9 (page 11, line 206).

12. "*Line 167: The difference you notice, is that big or small when compared to other CFD studies?*" Thank you for your comment. Within the IEA Task 29 Phase IV project, we compared codes with different fidelity levels against experimental data including several other CFD codes. Although the comparison done so far was without an imposed turbulence (but at the same flow conditions), all CFD codes agreed fairly well for all radial sections. In fact, our CFD simulations showed a slight improvement in accuracy as the turbulence was introduced.

13. "*Fig. 4: Also add what the differences are between the subplots a..f in the caption. I can deduce it is normal and tangential forces for different radial stations, but it took me a bit of time before I realised it.*" Thank you for the suggestion, it has been revised. Please consider that the pictures have been changed and further comments have been added.

14. "*Fig.5 : You could consider helping the reader in understanding this figure faster: dx is out of plane/flap, dy in-plane/edge, refer to the labels a..b, etc I assume that since you have deflections this it the coupled versions? This can be confusing since in the figure 6 you have a RIGID and COUPLED version, while nothing is specified in figure 5. You could argue this is obvious, but on the other hand it does avoid potentially confusing the reader.*" Thank you for the suggestion, we updated the pictures to make them more understandable.

15. "*Fig. 17 What is the unit of the y axis?*" The y-label has been added, thank you.

16. "*Line 187 What does this mean?*" The word figure was missing, and it has been corrected.

17. "*Fig. 18: The unit Knm referns to a load, not to a stress (which is in Pascal or N/m2)*" It has been corrected, thank you.

18. "*BMU case only considers 6 revolutions, while FMU/FMT 10 revolutions. Comparing cycling counts is only meaningful when both simulations have an equal length or revolutions*" That is correct, and it has been done like this,as explained also in comment 4.

**References**

[revised manuscript text omitted]

---

## Author Comment (AC3) · 10 Jun 2020

**Reply to comments by Reviewer Nr. 3**

Giorgia Guma on behalf of the authors
IAG, University of Stuttgart

June 10, 2020

The authors would like to thank the reviewer for his/her efforts and valuable comments. They are very much appreciated and incorporated into the revised paper.

In the present document the comments given by the 3rd reviewer are addressed consecutively. The following formatting is chosen:

- The reviewer comments are marked in blue and italic.

- The reply by the authors is in black color

- A marked-up manuscript is added. Changed section with regard to the comments by reviewer 3 are marked in green. Changed sections with regard to comments by more reviewers are marked in gray.

**General comments**

1. "*The overall subject of presenting a high fidelity FSI model is of considerable relevance for the research community as the turbines are continuously upscaled leading to more flexible designs where the aeroelastic effects become more and more important and require such FSI models to be truly analysed. The authors also mention this motivation in the introduction. However, a limitation of the paper content is that the simulations are carried out on a turbine design that is more than 20 years old and with a structural design that is much stiffer than designs that are more recent. Also the size of 2MW and a diameter of 80m for the chosen turbine is quite different from both recent reference turbine designs (e.g. the INNWIND and AVATAR reference rotors defined several years ago) in the research community as well as industrial designs that now have exceeded 10MW and 200m in diameter. The actual choice of the DANAERO turbine might be because detailed aerodynamic measurements are available for this turbine but the experimental data are only used in one case in the paper. So due to fact that the simulations are carried out on a relative stiff turbine design, some of the results and discussions of e.g. impact of flexibility on elastic torsion which are in the range of tenths of a degree, become somewhat theoretical without any real impact.*"

Thank you for your comment. Yes, you are right, this is a pretty old design, but the availability of high quality aerodynamic measurements data gives the IEA Task 29 participants the possibilty to validate and improve models and approaches of different levels of fidelity. These field measurements have been especially used within another subtask in order to validate the pure aerodynamic model that was then used within the simualtions of the aeroelastic task, and therefore for this paper. The main focus is to analyze the influence of resolved turbulence on the aeroelastic response of the turbine and therefore only a reference case has been validated with experimental results.

**Specific comments**

1. "*The present title indicates that the paper has a considerable focus and part on simulating turbulent inflow conditions. However, it is a limited part of the paper so it could be considered to change the title, e.g. to: A High Fidelity Simulation Tool for Aeroelastic Simulations of Wind Turbine*"
We think that modifying the title in this way would lead the reader to wrongly think that this FSI tool has been developed within this project. Although turbulence seems to be just a small part of the job, everything was made in order to get step by step to our last stop, turbulence respectively.

2. "*Introducing the turbine used for the simulations as ".. the DANAERO turbine ..." is not precise as many in the research community do not know that specific turbine. So it should be changed to a real description of the turbine, e.g. "a 2MW NM80 turbine with an 80m diameter rotor". Also the text: ".. specific case of the DANAERO experiment.." could be more informative, e.g. like: ".. of the inflow turbulence is analysed for a specific case in comparison with experimental data from the DANAERO experiment.""* Thank you for the suggestion, we have changed in $\boxed{\textbf{R3:S1}}$ (page 1, line 1) and $\boxed{\textbf{R3:S1bis}}$ (page 1, line 6).

3. "*Minor typos in line 22 and 29*" They have been corrected in $\boxed{\textbf{R3:S2}}$ (page 1, line 25) and $\boxed{\textbf{R3:S3}}$ (page 2, line 31).

4. "*Line 96-97 The description of the turbulence generation by body forces is very brief – please expand, e.g. over what axial distance are the body forces applied and how much does the turbulence decay down to the turbine ?*" We expanded it at $\boxed{\textbf{R3:S4}}$ (page 4, line 98).

5. "*Line 128 What is the actual FMT time simulation length and what did determine the length ?*" The FMT rigid has been calculated for a total of 20 revolutions, where 10 of them have been calculated in uniform inflow only. Afterwards, 10 revolutions more have been calculated in flexible state and rigid state at the same time, in order to be comparable. The DANAERO rotor presents a high induction, therefore it takes a long time for the wake to fully develop and to the loads to stabilze. In order to save computational costs we inject turbulence and let the structures deform, only after a cheaper simulation reached a low residuum, stable loads and a wake development long enough to avoid effects on the loads too. This informations has been added in $\boxed{\textbf{R3:S5}}$ (page 8, line 161).

6. "*Line 165 -167 The discussion of this case is very short – please expand*" Thank you, it has been expanded in $\boxed{\textbf{R3:S6}}$ (page 11, line 208). Please, consider that the pictures have been changed.

7. "*Line 165 -167 Why is not this case simulated with turbulence as the experimental data are shown, probably from several revolutions in turbulent inflow*" We have calculated a case with the same turbulence intensity but only in rigid, and for completeness it is now added to the same caption. Anyhow we did not calculate it taking into account flexibility because of the low turbulence intensity in the experiment, that would have not lead to a structural response much different from the one in uniform case. So, at the end, we did not do it in order to save computational cost, because we thought it was enough for the validation.

8. "*Line 173 - 174 The percentage values are shown with two decimals. They are probably based on the time trace values so maybe better just to show one decimal or how accurate are these values ?*" We agree that 2 decimals can be too brave, and therefore we changed it just to one, as you can see in $\boxed{\textbf{R3:S9}}$ (page 11, line 221).

9. "*Line 187 that Mx in 6 – please correct*" It has been corrected in $\boxed{\textbf{R1:S8}}$ (page 12, line 235)

10. "*Line 215 An increase in power for coupled case of 3.5% is quite much – is it a time varying flow due to separation in the region of 40-50% radius or what can be the causes?*" We analyzed the pressure distribution at 40% and 50% of the blade radius in order to verify it, and added the pictures in the text. We detected a maximum pressure difference between the two cases on the pressure side of 2.5%. This sums together with the increase of the rotor disk area (due to the stretching of the pre-bended blade) and can explain the evaluated increase in power. We added this in $\boxed{\textbf{R3:S11}}$ (page 15, line 284).

11. "*Line 248 " .. in FMT a peak appears close to the first flap bending eigenfrequency of the blade (f1 = 0.938Hz) that could lead to instabilities .. ". Please show the position of that frequence by a vertical line in the spectrum. As concerns instability – how can this lead to instability ?*" Thank you for your comment. Yes, you are right if the FFT of FMT tip deformations shows a peak close to f1, and FMU not, this peak is too small to be able to draw any conclusion about it. The sentence has been deleted and therefore no line on the spectrum was added, because not necessary anymore.

12. "*Line 282-285 Could numerical stabilities in the coupling generate the considerable number of small oscillations?*" Thank you for your comment. If there would be a stability problem of the coupling, it would show also in the deformations. These show for example in BMU, RMU and FMU to be really smooth, see fig. 6 and 16. The load oscillation occurs already in the rigid cases and it is a natural result of unsteady simulations where local inherently unsteady flow separation in the hub region causes slight load fluctuations. Therefore we do not think that it can be anyhow related to the stability of the coupling.

[revised manuscript text omitted]

---

## Author Response (AR2)

**Reply to comments by Reviewer Nr. 1**

Giorgia Guma on behalf of the authors
IAG, University of Stuttgart

September 18, 2020

The authors would like to thank the reviewer for his/her efforts and valuable comments in this second process of review. They are very much appreciated and incorporated into the revised paper.

In the present document the comments given by the 1st reviewer are addressed consecutively. The following formatting is chosen:

- The reviewer comments are marked in blue and italic.

- The reply by the authors is in black color

- A marked-up manuscript is added. Changed section with regard to the comments by reviewer 1 are marked in yellow.

**General comments**

1. "*A motivation for using CFD is given based on earlier found shortcomings of the BEM method. Still, would the main conclusion (influence of turbulence on fatigue loads dominates over flexibility) not also be obtainable using a BEM code? In other words I do see the added value in CFD for getting absolute values correct, but I question its additional value over BEM here in quantifying differences between computational settings such as flexibility. Perhaps the added value of this work lies more in validating the set-up of this detailed FSI model, allowing it to be used in applications where its significance is more clear?*"

Thank you for your comment. Yes you are right, a qualitative conclusion on the effects on the fatigue loading could have been done basing only on BEM. On the other hand, we think it was necessary to generate reference solutions, taking into account all physical effects together with a correct development of the turbulent structures in the flow field, to study the impact of the different interactions on unsteady loads. The scope of the paper was actually not the comparison between CFD and BEM, which calculations have been added under reviewer's suggestion, but we appreciated the proposal and decided to generate a model and include it. Of course we have now, additionally, a validated detailed FSI model that we also use for acoustic calculations (for example low frequency noise studies and trailing edge noise studies in IEA Task 39 and 29), where BEM cannot be used, but this is now out-of-topic. We revised and elaborated the conclusions according to your comment adding more aspects of the CFD-BEM comparison as you can see in $\boxed{\text{R1:G1}}$ (page 22, line 377).

2. "*A lot of observations are given in the text but often I am missing the main storyline and therefore the paper becomes difficult to follow, reducing its impact. A clear focus would improve the paper rather than discussing the results of a large number of different simulations.*"

Thank you for your observation. We agree that there are a lot of results given to separate the effects. We added therefore paragraph to the introduction in $\boxed{\textbf{R1:G2}}$ (page 2, line 42) describing the objectives and the procedure for clarification.

3. "*Captions do not always clearly indicate which operational condition (6.1, 9, 13m/s) or code (BEM/CFD) is used, which reduces readability.*"

Thank you for your comment, captions have been changed figures 6,7,8,10,11,12,13,16,17,19,21 and 22 following your suggestion.

**Specific comments**

1. "*Line 185 is it not Hendriks as mentioned in the reference?*"

Thank you for noticing it, we changed it in $\boxed{\textbf{R1:S1}}$ (page 9, line 192).

2. "*Fig. 9 Color coding BEM in agreement with CFD (fig. 7)?*"

Now that we changed the captions according to your comment, we changed the color coding to be in agreement between CFD and BEM as you can see for figure 9 (in agreement with figure 7) and 14 (in agreement with figure 11).

3. "*Fig. 9 Enormous tower effect in BEM, are we sure the input is correct? If not, what is the cause for this difference? Has this been oberved in previous literature? Or can you validate against the measurements of the DANAERO database?*"

Thank you for your comment. You are right, the tower effect computed is really strong. Comparing the BEM results in time with the measurements (in order to see the tower effect) is not really feasible because those have always been made under turbulent inflow conditions. This leads to high difficulties in recognizing the tower effect. It is also to be noticed that the DANAERO rotor features a prebend but no cone angle, that is usually used to avoid a large tower effect. AeroDyn v13 (the one coupled to SIMPACK) bases the calculation of the tower effect on the work of Bak ([Bak et al (2001)]) using a potential flow solution around a cylinder together with a tower dam model. The input for the BEM tower model bases on geometric properties (taken from the structural model, and therefore consistent), and a series of "Re vs Cd" properties. Those are rarely available when creating a tower model and in lack of this information, as it has been done also in other projects and suggested by Nrel, values have been taken from [Roshko, Anatol (1961)]. We also conducted calculations where we changed those to the standard values for a cylinder (with $c_D = 0.6$) to test its influence and almost nothing changed. In summary, Aerodynv13 does not allow to really influence the way the tower effect is calculated, altough additional options are available in Aerodynv15, that could not be tested with SIMPACK. It needs also to be noticed that this drop of $20-24\%$ results from the the consideration of a single blade. When the total rotor moment is then taken into account, the drop gets reduced to just a third of it, that is a value also observed for example in [Fu et al (2018)].

**References**

[Roshko, Anatol (1961)] Roshko, A.; Experiments on the flow past a circular cylinder at very high Reynolds number.; Journal of Fluid Mechanics, 10 (3). pp. 345-356. ISSN 0022-1120; 1961;

[Bak et al (2001)] Bak, C.; Aagaard Madsen, H.; Johansen, J. 2001.; Influence from blade-tower interaction on fatigue loads and dynamics (poster).; Wind energy for the new millennium.

Proceedings.; European wind energy conference and exhibition (EWEC '01). Copenhagen (DK), 2-6 Jul 2001; Helm, P.; Zervos, A. (eds.), (WIP Renewable Energies, München 2001) p. 394-397; 2001;

[Fu et al (2018)] Fu, L., Wei, Y. D., Fang, S., Tian, G., Zhou, X. J.; A wind energy generation replication method with wind shear and tower shadow effects; Advances in mechanical engineering, 10(3), 1687814018759216; 2018;

---

## Author Response (AR3)

**Reply to comments by Reviewer Nr. 1**

Giorgia Guma on behalf of the authors
IAG, University of Stuttgart

October 26, 2020

The authors would like to thank the editor for her efforts and valuable comments in this final process of review. They are very much appreciated and incorporated into the revised paper.

In the present document the comments given by the editor are addressed consecutively. The following formatting is chosen:

- The editor comments are marked in blue and italic.

- The reply by the authors is in black color

- A marked-up manuscript is added. Changed section with regard to the comments by the editor are marked in yellow.

**General comments**

1. "*The prior editor noted that the paper is missing a "story line". What is the main result? This is still not sufficiently addressed. Many of the detailed comparisons of the model in different configurations are not so interesting; please try to remedy this.*"

Thank you for your comment. In order to make the main result and the "story line" clearer we modified additionally the conclusions, see $\boxed{\text{R1:G1}}$ (page 23, line 382), $\boxed{\text{R1:G1a}}$ (page 24, line 413), addressing the main results this time additionally by the use of bullet points. The reason why we decided to compare the different configurations in detail is that, when computing those with CFD, computational costs vary enormously. It is therefore of interest, especially for the industry, to know limitations and differences within the high-fidelity modelling approaches, as addressed now additionally in the introduction at $\boxed{\text{R1:G1b}}$ (page 2, line 47).

2. "*And I agree that it might be a miss-interpretation writing: This has a direct effect on the DEL, being mostly affected by turbulence than flexibility and blade-tower passage together Because it needs a simulation with turbulence of a stiff and flexible structure. Although it's a stiff turbine I would guess that the DEL of the tower bottom moments and of blade moments are influenced by flexibility*"

Thank you for your comment. We understand that there might have been a bit of confusion and therefore we reformulated this sentence (and the conclusions) in $\boxed{\text{R1:G1}}$ (page 23, line 382), $\boxed{\text{R1:G1a}}$ (page 24, line 413). We simulated both the rigid and flexible turbine in turbulent inflow conditions and calculated the respective DEL based on the blade moments at the rotor center (tower bottom moments have not been considered). In this way we observed that the introduction of turbulence is the factor that mostly increases the DEL, independently of the flexibility, as shown in fig. 21 in the paper. Flexibility is, of course increasing the fatigue, but

much less in comparison to what turbulence does (comparison DEL "entire turbine in turbulent inflow FMT" vs "entire turbine in uniform inflow FMU"). This is what we wanted to explain with that sentence and that was probably a bit unclear.

3. "*Finally, please review and consider as a reference the work below with BEM and CFD simulations on the same turbine: Citation (APA): Madsen, H. A., Sørensen, N. N., Bak, C., Troldborg, N., Pirrung, G. (2018). Measured aerodynamic forces on a full scale 2MW turbine in comparison with EllipSys3D and HAWC2 simulations. Journal of Physics: Conference Series, 1037(2), [022011]. https://doi.org/10.1088/1742-6596/1037/2/022011 where the real objective of applying CFD in turbulent inflow in comparison with BEM is more clear than in the current study.*"

Thank you for your valuable suggestion. The citation has been added with a short comment in $\boxed{\textbf{R1:G3}}$ (page 7, line 153), within the description of the adopted BEM model.

---

## Author Response (AR4)

**Reply to comments by Chief Editor**

Giorgia Guma on behalf of the authors
IAG, University of Stuttgart

November 25, 2020

The authors would like to thank the editor for his efforts and valuable comments in this final process of review. They are very much appreciated and incorporated into the revised paper.

In the present document the comments given by the editor are addressed consecutively. The following formatting is chosen:

- The editor comments are marked in blue and italic.

- The reply by the authors is in black color

- A marked-up manuscript is added. Changed section with regard to the comments by the editor are marked in yellow.

**General comments**

1. "*Units should be in roman, not italic.*"

Thank you for your comment. All units have been changed from italic to roman, see **R1:G1a** (page 1, line 1), **R1:G1b** (page 3, line 68), **R1:G1b1** (page 1, line 14), **R1:G1c** (page 5, line 108), **R1:G1d** (page 7, line 165), **R1:G1e** (page 9, line 214), **R1:G1f** (page 13, line 267), **R1:G1g** (page 13, line 272), **R1:G1h** (page 17, line 304).

2. "*In the abstract you define abbreviations, even CFD which I think is good because if increases readability. Please also define BEM. Many people only or at least start to read the abstract and the conclusion. Therefore, please also define the abbreviations in the conclusion, again to increase readability.*"

Thank you for your valuable comment, we defined the abbreviations again in both abstract and conclusions, see **R1:G2a** (page 1, line 8), **R1:G2b** (page 22, line 382), **R1:G2c** (page 22, line 383), **R1:G2d** (page 22, line 388), **R1:G2e** (page 22, line 391), **R1:G2f** (page 23, line 394), **R1:G2g** (page 23, line 397), **R1:G2h** (page 23, line 401) and **R1:G2i** (page 23, line 405).

3. "*Figure 2 seems not to add much to the manuscript. It is hard to see the coordinate systems attached to the blade and also difficult to under the point of showing it. Could it be improved or removed?*"

Thank you for your valuable suggestion. Considering that it is only a beam model and therefore its full visualization consists of only points, we decided to follow your suggestion and delete it.

4. "*Various smaller things: References in parenthesis should be (Schepers, 2016), not (Schepers (2016)). Use citep or parencite if you use LaTeX. On figure 3 why are the same thing (?) called both $F_x$ and $F_N$? Figure 8: is Fx the same as $F_x$? Please be consistent. Additional formulations here and there could be improved, but the publisher will also help with that.*"

Thank you for your comment. We changed all citations in parenthesis using now the command citep as you suggested. And yes, exception for figure 4 where the loads are referred to the chord length, Fx, $F_x$ and $F_N$ (in figure 2) are the same thing. All figures related to the loads have now been consistently changed, see figures 2,6,7,8,10,11,18,19,20 and 22.